# Hydroxymethanesulfonate (HMS) Formation under Urban and Marine Atmosphere: role of aerosol ionic strength

Rongshuang Xu[a,b], Yu-Chi Lin[a,b], Siyu Bian[a,b], Feng Xie[a,b] and Yan-Lin Zhang[a,b*]

[a] School of Ecology and Applied Meteorology, Nanjing University of Information Science & Technology, Nanjing 210044, China
[b] Atmospheric Environment Center, International Joint Laboratory on Climate and Environment Change (ILCEC), Nanjing University of Information Science & Technology, Nanjing 210044, China

10    *Corresponding authors:*
Yan-Lin Zhang, dryanlinzhang@outlook.com; zhangyanlin@nuist.edu.cn

## Abstract

Hydroxymethanesulfonate (HMS) has emerged as a critical organosulfur species in ambient aerosols, yet the impact of aerosol properties, particularly ionic strength (IS), on the formation of HMS remains uncertain. Here, HMS levels in wintertime of urban Nanjing, China were quantified at $0.30\pm0.10$ μg m$^{-3}$, where the contribution of in-cloud formation likely carry minor significance due to the barrier resulted from stable stratification. Elevated HMS concentration was recorded during a Nanjing haze event resulted from enhanced HMS formation rates, which can be largely attributed to reduced IS levels on hazy days as the IS-dependent enhancement on HMS formation increased with decreasing IS within the continental IS range (6-20 mol kg$^{-1}$). This arises from the fact that elevated IS can boost HMS formation rate constants but also hinder the solubility of HMS precursor (SO$_2$) and its further dissociation. Consequently, the IS-dependent enhancement initially rose with increasing IS, peaking at 4 mol kg$^{-1}$ before declining. Additionally, for the first time, the particulate HMS level in marine atmosphere (Yellow Sea and Bohai Sea) were quantified at $0.05\pm0.01$ μg m$^{-3}$. Lower IS levels (2.0-6.0 mol kg$^{-1}$) observed for marine aerosols exhibited more pronounced enhancements on HMS formation, which can render the aerosol-phase HMS formation a process comparable to that in cloud/fog droplet under marine environments. Furthermore, the study highlights the significant impact of ambient humidity on aerosol IS (R=-0.89), suggesting the integration of ionic strength into chemical models to better represent the particulate sulfur chemistry, particularly in humid environments.

## Keyword

*Hydroxymethanesulfonate, aerosol ionic strength, aerosol acidity, urban haze, marine atmosphere*

# 1. Introduction

In recent decades, hydroxymethanesulfonate (HMS) has emerged as an important organosulfur (OS) species in polluted atmospheres, particularly in northern China (Moch et al., 2018; Song et al., 2019; Ma et al., 2020; Moch et al., 2020; Wei et al., 2020; Liu et al., 2021a; Chen et al., 2022; Wang et al., 2024), where peak daily-average HMS concentrations have been measured to reach 18.5 µg m$^{-3}$ (Ma et al., 2020). HMS is formed through the aqueous-phase reactions between dissolved sulfur dioxide ($SO_2$) and formaldehyde (HCHO) (Boyce and Hoffmann, 1984; Dixon, 1992; Zhang et al., 2023). Atmospheric chemical models provided additionally insights into the prevalence and significance of atmospheric HMS, demonstrating that HMS can account for 10% of global aerosol sulfur in continental surface air and over 25% in urban polluted areas during winter season (Moch et al., 2020; Song et al., 2021; Wang et al., 2024). Regional model simulation also proposed that with the rising concentrations of HCHO and the heightened marine productivity of DMS under climate change (Kim et al., 2018), HMS can play an progressive importance in coastal and marine aerosols (Zhao et al., 2024), potentially impacting their CCN activities (Zhang et al., 2024). However, validation of these conclusions requires observational evidence regarding the abundance of HMS in pristine costal and even marine atmospheres.

Several measurement-based studies have suggested that meteorological conditions, specifically lower temperatures (T) and elevated relative humidity (RH) resulting in increased atmospheric liquid water content (LWC), can facilitate the dissolution of gaseous precursors, thereby accelerating the aqueous-phase HMS formation (Ma et al., 2020; Wei et al., 2020; Chen et al., 2022). For instance, the HMS production rate is expected to increase linearly with the LWC (**Eqn.2**). In addition, other physicochemical properties of the aqueous medium (i.e., cloud/fog droplet, aerosol bulk water) including acidity (pH) and ionic strength (IS) has more pronounced impact on the HMS production (Song et al., 2019; Song et al., 2021; Zhang et al., 2023). Aqueous acidity can influence the HMS formation by affecting the solubility of gas-phase $SO_2$ and equilibrium distribution among S(IV) (i.e., $SO_2 \cdot H_2O$, $HSO_3^-$ and $SO_3^{2-}$) (Berglen et al., 2004) for subsequent reaction with $HCHO_{(aq)}$. A pH range of 4 to 6 has been proposed as a favorable acidity condition for HMS formation, above which HMS can becomes unstable and decompose into $SO_3^{2-}$ and $HCHO_{(aq)}$ (Boyce et al., 1984; Olson and Hoffmann, 1986; Moch et al., 2018; Song et al., 2019). Theoretically, the HMS formation rate can decrease by a factor of 100 from pH 6 to 5 assuming a constant water content. Field observations even observed an inverse correlation between HMS concentration with aerosol pH (Scheinhardt et al., 2014; Zhang et al., 2024). This suggests that in realistic atmospheric environments, there are likely additional factors that play significant roles in governing the HMS formation.

Recently, laboratory study found that the higher ionic strength in aerosol water (i.e., $1 \leq IS \leq 11$ mol kg$^{-1}$) can significantly amplify the rate constants for HMS formation (reaction **R1** and **R2**) by 2-3 orders of magnitude compared to dilute solution (IS<10$^{-2}$

mol kg$^{-1}$) (Zhang et al., 2023). Building upon this finding, a model study revealed that the formation of HMS in aqueous aerosols can account for approximately one-third of observed HMS levels during the winter season in Beijing, and also suggest that in certain scenarios, the enhancing effect of IS can counterbalance the constraints imposed by low aerosol water content and pH values (Wang et al., 2024). Nonetheless, the exact extent and scope of this compensatory mechanisms remain ambiguous. Besides, ambient aerosols properties can be highly distinct under diverse atmospheres, governed by the environmental conditions and aerosol compositions (Herrmann et al., 2015). For instance, the aerosol LWC can exhibit a wide range, from tens of $\mu g\ m^{-3}$ in the marine atmosphere to hundreds of $\mu g\ m^{-3}$ under severe Asia haze (Gopinath et al., 2022). Aerosol particles in urban regions are commonly acidic, with pH levels spanning from 3 to 5 (Liu et al., 2021b). During haze pollution, pH values can fluctuate due to factors such as increased ammonia (NH$_3$) emissions or the secondary formation of sulfate, nitrate, and organic acids, potentially resulting in either an increase or decrease in aerosol pH (Shi et al., 2019; Ruan et al., 2022). Freshly emitted sea spray aerosols can undergo rapid acidification, achieving pH ranges of 3.5 to 4.7 (Yu et al., 2023), fostering conditions conducive to HMS formation. The aerosol ionic strength can also vary significantly, ranging from 1 to 20 mol kg$^{-1}$, influenced by the aerosol composition of ionic species and the RH conditions (Herrmann et al., 2015; Mekic and Gligorovski, 2021). For example, thermodynamic models have indicated a notably higher IS level for urban aerosols (~20 mol kg$^{-1}$) (Hennigan et al., 2015) compared to marine aerosols (IS = 6.1 mol kg$^{-1}$) (Sander and Crutzen, 1996), likely attributed to the humid and pristine conditions typically found in marine atmospheres. In addition, above properties typically intricately interdependent within atmospheric aerosols. Thus, a comprehensive evaluation of the role of aerosol characteristics in atmospheric HMS formation, particularly the interactions between aerosol ionic strength and potentially elevated acidity across different atmospheres, is highly desired.

In sight of this, this study conducted field measurements for particulate HMS level in a continental city (Nanjing, China). As a comparison, marine aerosols were also collected during a cruise across the Yellow Sea (YS) and Bohai Sea (BS), China. The HMS abundance in collected PM$_{2.5}$ samples were quantified using ion chromatography (IC). Aerosol properties were estimated by thermodynamic models. Constrained by precursors' concentrations, the potential HMS production rates were also estimated and the role of aerosol properties in HMS formation under urban and marine environments were also examined. Results from this work shall provide improved insights on particulate sulfur chemistry within diverse atmospheric conditions.

## 2. Material and Method

### 2.1 Atmospheric measurements in urban and marine environments

Urban aerosols and gaseous pollutants were measured in Nanjing in the winter of 2023 (**Figure S1** in the Supplement **Section S1**). From 17[th] December, 2023 to 10[th] January, 2024, PM$_{2.5}$ samples were systematically collected for a duration of 23 hours

daily. The daily averaged concentration of HMS together with other water-soluble inorganic ions within $PM_{2.5}$ were determined by Ion chromatograph (Dionex Corp., CA, US). Given that free S(IV) or other S(IV) species such as aldehyde-S(IV) adducts may be misidentified as HMS during the IC analysis (Ma et al., 2020; Dingilian et al., 2024), hydrogen peroxide ($H_2O_2$) was employed in analysis process to specifically isolate HMS (Dingilian et al., 2024). The results suggested the influence of free S(IV) or other S(IV) species on HMS measurement was negligible, as elaborated in the Supplementary Information (**Section S1**). Carbonaceous components including organic carbon (OC), elemental carbon (EC) and water-soluble organic carbon (WSOC) were measured using a sunset OC/EC analyzer (Sunset Laboratory Inc.) and a total organic carbon analyzer (TOC-L, Shimadzu, Kyoto, Japan), respectively. The levels of gaseous pollutants (i.e., $SO_2$, CO, $O_3$) in close proximity to our sampling location were obtained from National urban air quality and real-time publishing platforms (http://106.37.208.233:20035/). The hourly level of formaldehyde (HCHO) was retrieved based on the Multi-Axis Differential Optical Absorption Spectroscopy (MAX-DOAS) data (**Section S2**). To calculate the pH values in aerosols, three semi-volatile gases ($NH_3$, $HNO_3$, and HCl) were also monitored at 30-minute intervals using a Monitor for AeRosols and GAses (MARGA; Metrohm Ltd., Switzerland). Meteorological data (i.e., RH, temperature, wind speed, visibility, etc.) and backward trajectories (**Section S3**) were also obtained. In addition, daily $PM_{2.5}$ samples, gas-phase $SO_2$ samples and meteorological data were collected onboard during the open research cruise NORC2024-01 conducted in the spring of 2024 (from April 13th to April 25th). Composition of marine aerosols and ambient $SO_2$ levels were also characterized. Details of the sample collection and chemical analysis were described in **Section S1**.

**2.2 Ambient aerosol properties estimation**

Recognizing the influence of organic components (OA), especially the WSOC on aerosol properties, here we integrated two established thermodynamic models, namely ISORROPIA II and the Aerosol Inorganic-Organic Mixtures Functional groups Activity Coefficient (AIOMFAC) models, to estimate aerosol liquid water content (ALWC), pH, and ionic strength (IS), following the methods delineated in previous work (Battaglia et al., 2019). The ISORROPIA II model (Fountoukis and Nenes, 2007) has gained widespread adoption in estimating aerosol characteristics for atmospheric particles (Nah et al., 2019; Zhang et al., 2024), which considers the thermodynamics of inorganic ions present in aerosol and their equilibrium with gas-phase $HNO_3$, $NH_3$, HCl and $H_2O$, without the consideration of organic constituents. On the other hand, AIOMFAC model (http://www.aiomfac.caltech.edu/model.html, last access: 16 September 2024) offers the most extensive consideration of the organic-inorganic interaction but does not solve the equilibrium partitioning calculations. Thus, this study integrated both models to deliver a more comprehensive assessment of aerosol properties. In brief, inorganic PM composition ($SO_4^{2-}$, $NO_3^-$, $Cl^-$, $NH_4^+$, $Ca^{2+}$, $K^+$ and $Mg^{2+}$) and gaseous $HNO_3$, $NH_3$, HCl data observed in urban Nanjing were input into ISORROPIA-II under "Forward" mode and "metastable" state to derive equilibrium concentrations of ALWC and all ionic species present under specified temperature (T)

and RH conditions. Subsequently, organic components were incorporated into the inorganic matrix at the same T and RH settings, and the aerosol compositions were simulated using AIOMFAC to calculated the ALWC, pH, and IS. For marine aerosols, we adopted a particle to particle + gas partitioning fraction of $NH_4^+$ ($\varepsilon$) of 0.5, drawing from prior observations conducted in the Bohai Sea in May (Wang et al., 2022a) in light of the absence of gaseous data. Details regarding the model setup and the comparison between model outputs with inorganic-only simulations using ISORROPIA II can be found in **Section S4**.

## 2.3 Kinetical description for HMS production

In the atmosphere, the formation of HMS requires its gas precursors ($SO_2$ and HCHO) firstly dissolve into the aqueous water. Dissolved S(IV) (i.e., $HSO_3^-$ and $SO_3^{2-}$) can subsequently react with $HCHO_{(aq)}$, leading to the HMS formation:

$$HSO_3^- + HCHO_{(aq)} \overset{k_1}{\leftrightarrow} HOCH_2SO_3^- \qquad \textbf{(R1)}$$

$$SO_3^{2-} + HCHO_{(aq)} + H_2O \overset{k_2}{\leftrightarrow} HOCH_2SO_3^- + OH^- \qquad \textbf{(R2)}$$

These reactions are reversible, and the first-order rate constant of HMS decomposition have been measured in previous studies showing the decomposition of HMS was so slow that HMS levels are predominantly controlled by formation kinetics in the typical pH range of cloud/fog and aerosol water (pH < 6) (Boyce et al., 1984; Deister et al., 1986; Kok et al., 1986; Song et al., 2021). The $k_1$ and $k_2$ represent the second-order rate constants of **R1** and **R2,** respectively. where $k_2$ is ~ 4 orders of magnitudes larger than $k_1$. Thus, the HMS production can be highly sensitive to aerosol acidity, which governs the solubility and distribution of S(IV) species. The $R_{HMS}$ (M s$^{-1}$) can be represented by:

$$R_{HMS} = (k_1 \times [HSO_3^-] + k_2 \times [SO_3^{2-}]) \times [HCHO_{(aq)}] \quad \textbf{(Eqn.1)}$$

Furthermore, the HMS formation rate ($P_{HMS}$) in ambient aqueous medium (i.e., aerosol water) was calculated in the unit of $\mu g\ m^{-3}\ h^{-1}$:

$$P_{HMS} = R_{HMS} \times ALWC \times 3600 \times M_{HMS} \quad \textbf{(Eqn.2)}$$

where ALWC were estimated from thermodynamic models and $M_{HMS}$ is the molar mass of HMS ($M_{HMS}$=111 g mol$^{-1}$). Here we considered the impact of aerosol ionic strength on HMS formation by determining the IS-dependent $SO_2$ solubility, dissociation of $H_2SO_3$ as well as the rate constants ($k_1$ and $k_2$) for **R1** and **R2**, as detailed in **Section S5**.

## 3. Results and Discussion

### 3.1 Detection of sulfur-containing species in urban and marine atmosphere

**Figure 1** shows the observations of particulate sulfur-containing species (i.e., HMS and sulfate) and $SO_2$ level in urban and marine atmospheres, together with PM$_{2.5}$ compositions, temperature and relative humidity conditions. In urban Nanjing, a total of 25 daily PM$_{2.5}$ samples were collected from December 17[th], 2023 to January 10[th], 2024. The HMS concentrations ranged from 0.18 to 0.56 $\mu g\ m^{-3}$ (averaging at

0.30±0.10 μg m$^{-3}$), with corresponding sulfate concentrations varying from 2.06 to 18.03 μg m$^{-3}$ (7.86±4.54 μg m$^{-3}$). The HMS/sulfate molar ratios fell between 2.5% and 8.5%, averaging at 4.6±1.7%. During the cruise, thirteen marine PM$_{2.5}$ samples and SO$_2$ samples were synchronously collected from April 13$^{th}$ to April 25$^{th}$, 2024. Due to the sampler malfunction and limited sampling duration amid adverse weather conditions, HMS was only detected in 6 daily marine aerosol samples, with lower concentration ranging from 0.03 to 0.07 μg m$^{-3}$ (0.05±0.01 μg m$^{-3}$). Despite the restricted sample number, this study provided the first observational evidence for HMS level in marine atmosphere, roughly aligned with previous model-estimated HMS level in marine atmosphere (0.03-0.09 μg m$^{-3}$) (Zhao et al., 2024). The ambient concentrations of non-sea-salt sulfate (nss-SO$_4^{2-}$) were from 1.50 to 2.70 μg m$^{-3}$ (2.09±0.46 μg m$^{-3}$). This notably lower particulate sulfur content in marine aerosols was likely attributed to the lower gas-phase precursor concentration, evidenced by an average SO$_2$ level of 0.82±0.47 ppb, compared to the SO$_2$ level in urban Nanjing of 2.2±0.7 ppb. And slightly lower HMS/sulfate molar ratios (3.0±1.1%) were observed in marine atmosphere considering the biogenic contribution to nss-SO$_4^{2-}$ in spring season (Zhang et al., 2013).

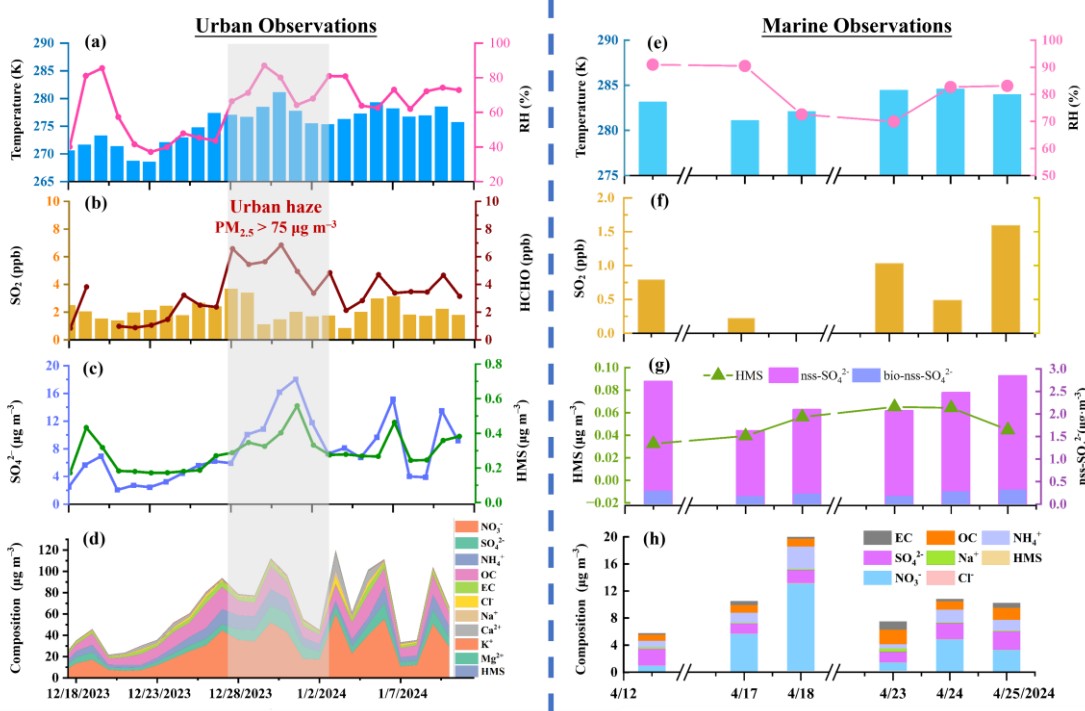

**Figure 1.** Left panel (Urban Observations): Time series of atmospheric measurements in urban Nanjing, including temperature and relative humidity (a); SO$_2$ and HCHO levels (b); HMS and sulfate concentration (c); PM$_{2.5}$ composition (d). The gray shade area highlighted a 7-days' haze episode (December 27$^{th}$, 2023 to January 2$^{nd}$, 2024) with PM$_{2.5}$ mass concentration exceeding 75 μg m$^{-3}$. Other ancillary measurements in urban atmosphere including wind speed and wind direction, PM$_{2.5}$ mass concentration, O$_3$, NH$_3$ level, fog occurrence and cloud water content can be found in **Figure S2**. Right panel (Marine Observations): Atmospheric measurements during marine cruise, including temperature and relative humidity (e); SO$_2$ (f), HMS and sulfate level where the biogenic nss-sulfate (bio-nss-sulfate) was calculated based on previously measured bio-nss-sulfate/nss-sulfate ratio of 0.14 (Yang et al., 2015) (g); PM$_{2.5}$ composition (h).

**3.2 Elevated HMS level during a haze pollution period in urban Nanjing**

In addition, a seven-day haze pollution event (from December 27[th], 2023 to January 2[nd], 2024) was observed in urban Nanjing, where a notably elevated levels of HMS and sulfate were observed (P<0.05). Throughout the haze event (PM$_{2.5}$ = 114.3±18.0 μg m$^{-3}$), the average concentrations of HMS and sulfate were 0.36±0.09 and 11.4±4.0 μg m$^{-3}$, respectively, with HMS/sulfate molar ratio of 3.5±0.7% (**Table S1**). It is noted that even under hazy days, the HMS level in our observations were significantly lower compared to those reported in Northern China during severe winter haze episode (averaged at 4-7 μg m$^{-3}$) (Ma et al., 2020; Liu et al., 2021a; Chen et al., 2022; Wang et al., 2024). Such divergence can be largely attributed to the contribution of HMS formed in cloud/fog processes, as evidenced by a previous study reporting comparable levels of HMS (< 1 μg m$^{-3}$) in Beijing to this work in the absence of cloud/fog, while peak HMS levels of 18.5 μg m$^{-3}$ was observed during fog processes (Ma et al., 2020). During our sampling period in Nanjing, there were no prolonged fog events (RH>90% and Visibility<1 km) lasting over two hours (**Figure S2e**). Additionally, previous study has emphasized that although in-cloud formation processes can be much faster compared to aerosol water given its large water content, their contribution to the near-surface aerosol composition can be negligible during China's winter haze due to the accompanied temperature inversions and high atmospheric stability so that the weak vertical exchange impeded the transportation of gas precursors emitted near surface to high altitudes and the chemicals produced in the high-altitude cloud could not be easily transported to the ground (Wang et al., 2022b). Here, we utilized the vertical temperature profiles retrieved from http://weather.uwyo.edu/upperair/sounding.html to identify temperature inversion within atmospheric boundary layer in the winter of Nanjing. As shown in **Figure S3**, a predominant occurrence of temperature inversions was noted across our observation days, characterized by inversion layers extending from the surface to altitudes of 2000 m, particularly evident on hazy days. Furthermore, days lacking temperature inversions (from December 19[th] to December 25[th], 2023) were associated with negligible cloud water content (**Figure S2f**), which were obtained from MERRA-2 (Modern-Era Retrospective analysis for Research and Applications, Version 2) (Gelaro et al., 2017). In line with Ma et al.'s work (2020), we also detected HMS within PM$_{2.5}$ samples during 7 of non-cloud days. Therefore, given the protective effects of temperature inversion and low wind speeds (**Figure S2a**) as well as reduced cloud water content prevalent during our observations, the contribution of in-cloud/fog HMS formation to observed particulate HMS levels was likely insignificant and the notably lower HMS level further suggest that the aerosol bulk water might serve as a predominant medium for HMS formation during our observations.

Other factors such as the level of precursors and atmospheric oxidants may also contribute to this divergence. For instance, during the winter haze in Beijing, the average level of HMS, sulfate were reported to be 4 μg m$^{-3}$ and 45 μg m$^{-3}$, respectively, with HMS/sulfate of 10%, and the gaseous SO$_2$ and HCHO concentrations were recorded at 12 and 20 ppb, respectively (Ma et al., 2020). This study observed

significantly lower levels of $SO_2$ (2.2±0.7 ppb) and HCHO (5.4±1.1 ppb) during the haze event. On the other hand, the reduced HMS/sulfate ratio could potentially be elucidated by the higher $O_3$ level in this work (43.3±4.7 ppb) compared to Beijing (5 ppb) as low atmospheric oxidation level could encourage more $SO_2$ to participate in the formation of HMS rather than sulfate (Song et al., 2019; Ma et al., 2020; Campbell et al., 2022).

In addition, the $SO_2$ concentration exhibited minimal variations throughout the entire sampling period, while the HCHO concentrations (ranging from 0.88 to 7.11 ppb where data are available) increased during $PM_{2.5}$ pollution episode (**Figure 1b**). HMS level strongly correlated with HCHO with Pearson correlation coefficient, R, of 0.61, but shows no correlation with $SO_2$ (**Figure S4**). However, peaks in HCHO levels did not consistently coincide with peaks in HMS concentrations, suggesting that other factors beyond these specific precursors, such as meteorological condition and aerosol physicochemical properties, might exert a more significant influence on HMS formation (Campbell et al., 2022). Thus, correlation analysis between HMS level and other factors including PM compositions, gas pollutants as well as meteorological conditions were conducted, in order to delve deeper into the potential drivers behind HMS formation in the winter of Nanjing (**Figure S4**). The HMS concentration exhibited strong correlations with sulfate and $PM_{2.5}$ level, with R values of 0.87 and 0.67, respectively. The strong HMS-sulfate correlation can be partially explained by the concurrent generation of HMS and sulfate, given that both species were primarily formed through aqueous-phase S(IV) chemistry. A prior study has highlighted a strong correlation between HMS concentration and organic aerosol (OA) in Fairbanks, where OA can dominate the ALWC due to its substantial contribution to $PM_{2.5}$ mass (up to 75%) and moderate hygroscopicity (Campbell et al., 2022). In this work, organic aerosols were also identified as crucial components in both urban and marine aerosols (**Figure S5**). In urban aerosols, organic aerosol mass (OA=1.6×OC) (Turpin and Lim, 2001) contributed between 10.4% to 43.6% to $PM_{2.5}$, where water-soluble organic carbon (WSOC) accounted for an average of 62%. For marine aerosols, the contribution of organic OA stood at 21.4±6.4%, with nearly two-thirds constituting WSOC species. Organic components in urban aerosols, particularly WSOC, also exhibited a strong correlation with sulfate and HMS levels during our observations (**Figure S4**), probably indicating their likely involvement in aqueous sulfur chemistry processes.

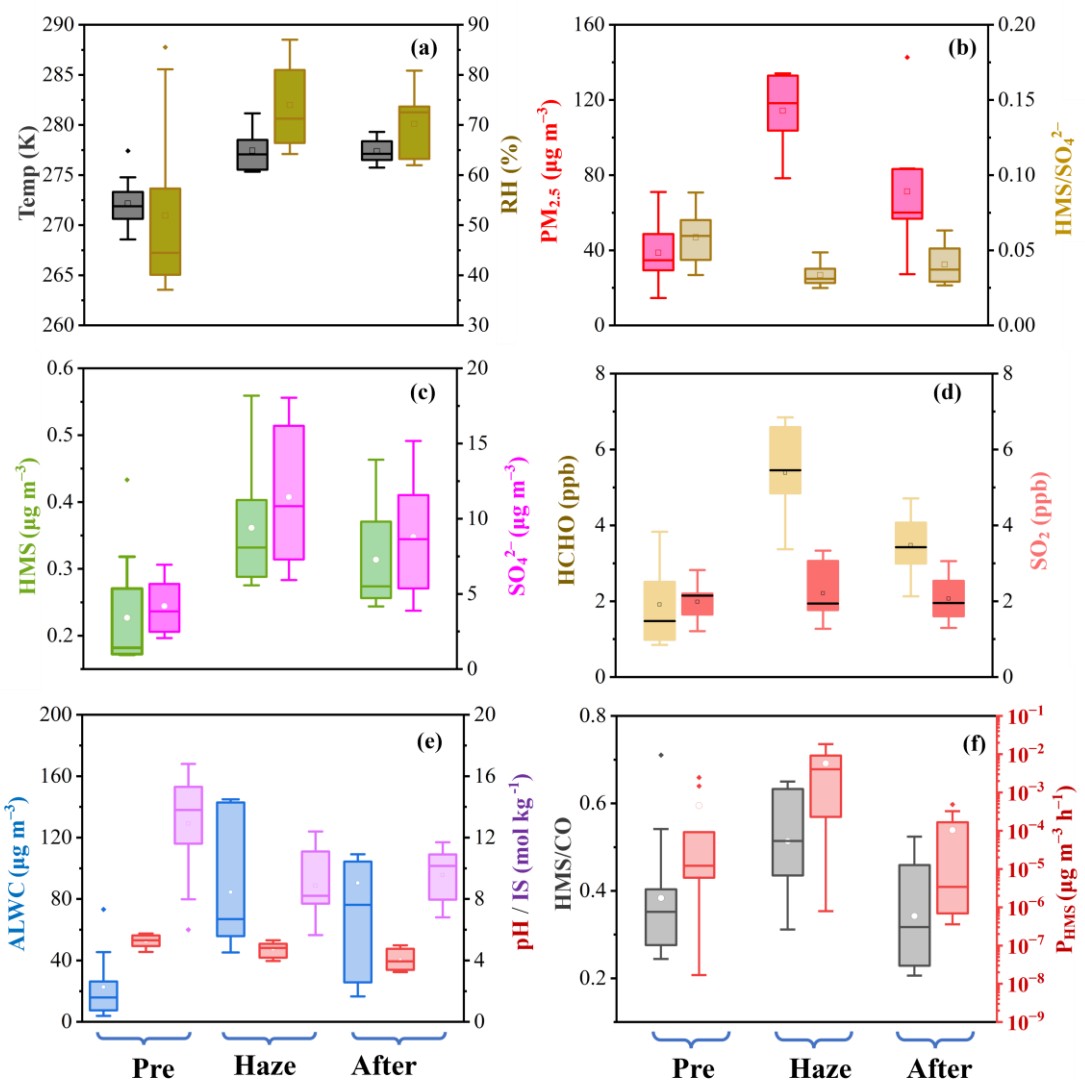

**Figure 2.** The averaged atmospheric characteristics under different pollution scenarios in urban Nanjing (i.e., pre-haze, haze and after) including: temperature and RH (a); $PM_{2.5}$ and HMS/sulfate molar ratio (b); HMS and sulfate concentration (c); gas precursors levels (d); aerosol properties such as liquid water content (ALWC), pH and ionic strength (IS) (e); and comparison (f) between HMS/CO ratio and HMS formation rates ($P_{HMS}$). The detailed values pertaining to the parameters mentioned above are provided in **Table S1**.

Here, notable correlations (R=0.75) between HMS and RH (ranging from 37% to 87%) were observed, aligning with prior research as elevated humidity can enhance the particle water uptake and subsequent gas dissolution. Additionally, HMS levels exhibited a significant correlation with temperature (R=0.60), contrary to findings in Fairbanks (Campbell et al., 2022), where lower temperatures were linked to enhanced solubility of $SO_2$ and HCHO, thereby facilitating HMS formation. It can be explained by that in this study higher temperatures were generally observed during haze days, typically associated with conditions that favoring HMS formation such as increased humidity and HCHO levels (**Figure 2**), ultimately resulting in a positive correlation

between HMS concentration and temperature. Furthermore, no significant correlation was found between HMS and wind speed (WS). The backward trajectories analysis also indicated that the air masses during haze episode predominantly originated from proximate regions (75%), while a minor proportion (25%) was transported from the eastern marine area (**Figure S6**). Collectively, these findings might suggest that observed particulate HMS level in Nanjing be largely formed on-site in aerosol aqueous water rather than being transported after its formation.

**3.3 Role of aerosol properties in HMS formation**

Therefore, the role of aerosol properties on the HMS formation was further evaluated based on ALWC, pH and ionic strength (IS) estimated by thermodynamic models (**Figure S7**). For urban aerosols, the estimated ALWC ranged from 4.5 μg m$^{-3}$ to 144.9 μg m$^{-3}$ throughout the entire observation period, with elevated ALWC levels during haze pollution, averaging at 84.5±38.4 μg m$^{-3}$ (**Figure 2, Table S1**). The variations in ALWC could be explained by the differences in RH and aerosol compositions as indicated by their strong correlations. The range of aerosol pH estimations fell between 3.2 and 5.8, a range known to be favorable for HMS formation (Song et al., 2019; Moch et al., 2020; Song et al., 2021). Notably, during haze pollution, the aerosol pH levels were estimated to be half-unit lower (4.7±0.5) than those under clean days (5.2±0.4) (**Figure 2**). This trend has been reported in Nanjing (Sha et al., 2019) and other regions such as Beijing (Ruan et al., 2022) and Sichuan Basin (Fu et al., 2024), a pattern frequently linked to enhanced sulfuric acid and nitric acid formation. The RH conditions and sulfate levels were identified as pivotal factors influencing pH fluctuations in this study based on a Random Forest model combined with Shapley additive explanations (SHAP) analysis (Zhang et al., 2025) (as detailed in **Section S4**). We also estimated the aerosol ionic strength, ranging from 5.6 to 16.8 mol kg$^{-1}$, falling within the typical IS range for continental aerosols (Herrmann et al., 2015). Consistent with prior research (Song et al., 2018), we noted higher ionic strength under lower humidity conditions. A pronounced negative correlation between IS with RH (R=-0.89) was identified, even in situations characterized by heightened PM$_{2.5}$ levels (**Figure 3a)**, which can be attributed to that under high humidity conditions, an increased formation of hygroscopic species can ultimately lead to a more diluted aerosol bulk solution. Specifically, with RH increasing by 10%, the aerosol ionic strength can decrease by around 2 units. This phenomenon accounted for the lower IS levels observed during pollution event (8.8±2.1 mol kg$^{-1}$, 74±8%) compared to the dry and clean days before (12.9±3.3 mol kg$^{-1}$, 52±17%). While acknowledging that the our regression between RH and IS may not be universally generalizable to other regions, our findings are consistent with earlier field and model research indicating a decline in aqueous ionic concentrations on humid haze days due to the rapid rise in ALWC (Song et al., 2018; Shen et al., 2024a; Wang et al., 2024).

HMS levels displayed a notably positive correlation with ALWC (R=0.67), in agreement with prior studies (Ma et al., 2020; Campbell et al., 2022), but demonstrated an inverse relationship with pH variations and negatively correlated with ionic strength.

The inverse relation between HMS level and pH has been reported in previous studies (Scheinhardt et al., 2014; Zhang et al., 2024) but not been fully explained as it contrasted to the established HMS formation chemistry where the HMS formation rates ($P_{HMS}$) can decreased theoretically by 10 to 100 times with 1-unit reduction in aerosol pH. Moreover, the negative correlation with ionic strength appeared to contradict earlier laboratory investigation that showed a continuous increase in reaction rate constants for **R1** and **R2** with rising IS from <1 mol kg$^{-1}$ (mimicking cloud/fog condition) to 11 mol kg$^{-1}$ (mimicking urban aerosol condition) (Zhang et al., 2023). This could be explained by dual impacts of ionic strength, which can not only enhance the reaction rate constants for **R1** and **R2** (Zhang et al., 2023) but also hinder the solubility of $SO_2$ and influence the dissociation constants of $H_2SO_3$ (Millero et al., 1989) (**Figure S8**). Here the IS-dependent influence on the HMS formation rate was characterized in terms of an enhancement factor (EF), calculated as EF=$P_{HMS}$/$P_{HMS, dilute}$. The $P_{HMS, dilute}$ was calculated using the classical parameters obtained in dilute solution (i.e., IS < 1 mol kg$^{-1}$), mimicking the condition of cloud/fog droplets (Seinfeld and Pandis, 2016). Thus, EF>1 indicated that the ionic strength level resulted in a faster HMS formation rate compared to that within cloud/fog droplets. As shown in **Figure 3b**, the IS-dependent enhancement on $P_{HMS}$ exhibited a discontinuous response to variations in IS, showcasing peak enhancement at an IS level of approximately 4 mol kg$^{-1}$ with EF>3000 at pH of 4, beyond which the EF gradually diminished. However, when IS surpassed 16.5 mol kg$^{-1}$, the EF dropped below 1.0, indicating that higher IS level may hinder rather than promote HMS formation. In urban Nanjing, the aerosol ionic strength (6-20 mol kg$^{-1}$) fell in the range where increasing IS level impeding the HMS formation and every 2-unit reduction in ionic strength resulted from 10% increase of humidity level can in turn accelerated the $P_{HMS}$ by around 10 times, which also explained the observed negative correlation between HMS level and aerosol IS (R=-0.62).

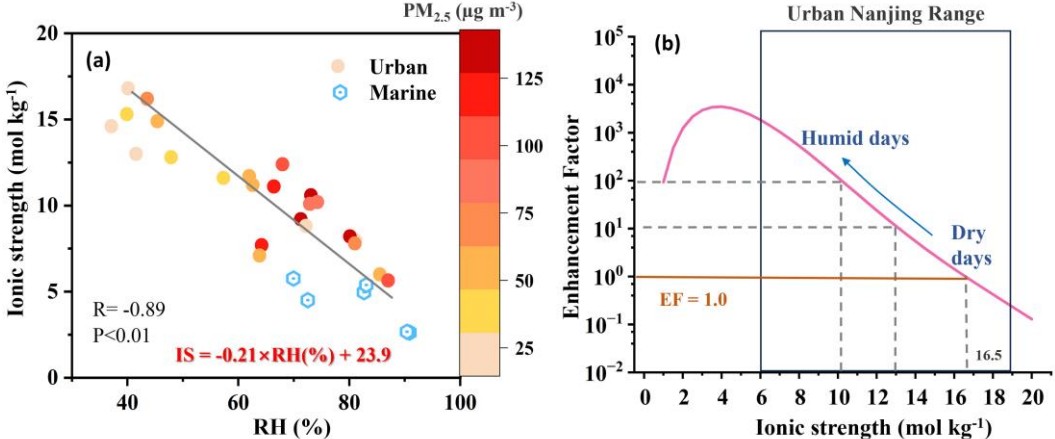

**Figure 3**. (a) The correlation between aerosol ionic strength and relative humidity level; (b) The enhancement factor (EF) exerts by ionic strength on the HMS formation rate, calculated as EF=$P_{HMS}$/$P_{HMS, dilute}$ at pH of 4. $P_{HMS, dilute}$ was calculated using the parameter proposed for dilute cloud water when the ionic strength was smaller than 1 mol kg$^{-1}$. Beyond this threshold (IS≥1 mol kg$^{-1}$), the $P_{HMS}$ was calculated following the equations given in **Table S3**.

The daily averaged steady-state $P_{HMS}$ throughout the sampling period in Nanjing was further determined (as detailed in **Section S5**) to explore the potential role of aerosol properties such as acidity and ionic strength in ambient HMS formation. Given that carbon monoxide (CO) was usually considered to be an inert chemical species during rapid haze formation, with its increase often interpreted as indicative of the accumulation of primary pollutants in the shallower boundary layer (Williams et al., 2016), the ratio of HMS to CO (HMS/CO) was calculated to better represent the secondary formation of ambient HMS (**Figure 2f**). During our study period, enhanced HMS formation was noted during hazy days as evident by the higher HMS/CO ratio (0.51±0.11). The $P_{HMS}$ estimations exhibited a good correlation with HMS/CO (R=0.57) (**Figure S9a**), and can roughly capture the diurnal variations of HMS/CO during pollution period. Besides, elevated HMS formation rates were found during haze event with averaged $P_{HMS}$ of $5.8\pm5.9 \times 10^{-3}$ µg m$^{-3}$ h$^{-1}$, an order of magnitude higher than that during the clean periods ($4.5\pm8.4\times 10^{-4}$ µg m$^{-3}$ h$^{-1}$) (**Figure 2f**). Certainly, when the ionic strength effect in aerosol water was not considered, daily $P_{HMS}$ was generally calculated to be one to two orders of magnitude lower (**Figure S9b**). Noteworthy, even with a 4-fold increase in ALWC coupled with a 2-fold increase in HCHO levels, these factors cannot completely counterbalance the potential 10-fold reduction in HMS formation rates resulting from the decreased aerosol pH, leading to slower HMS formation rates ($1.2\pm1.7\times 10^{-4}$ µg m$^{-3}$ h$^{-1}$) during hazy days compared to clean days ($3.3\pm5.3\times 10^{-4}$ µg m$^{-3}$ h$^{-1}$) (**Figure S9c**), failing to explain the higher HMS/CO ratio and HMS levels. Similar discrepancies have also been highlighted in previous studies where estimated HMS formation rates, without the consideration of high IS level in aerosol water, inadequately represented the ambient HMS concentrations and their temporal fluctuations (Ma et al., 2020; Campbell et al., 2022; Zhao et al., 2024). For instance, prior study has reported a $P_{HMS}$ of $2.6 \times 10^{-2}$ µg m$^{-3}$ h$^{-1}$, which was estimated using parameters (i.e., SO$_2$ solubility, $k_1$ and $k_2$) obtained in dilute solution and thus failed to represent the significant HMS level in Beijing (up to 18.5 µg m$^{-3}$) (Ma et al., 2020). These results raised a possibility that the HMS formation rate could be largely underestimated without considering high ionic strength level in aerosol water. During our observation, we found that it was the reduction in aerosol ionic strength on humid and polluted days led to more pronounced enhancement in HMS formation. This ultimately led to a nearly 10-fold rise in $P_{HMS}$ during haze episode compared to dry and clean days (**Figure S9d**), contributing to the elevated HMS level and HMS/CO ratio during haze event.

It was noted that the IS-dependent impacts on SO$_2$ solubility and H$_2$SO$_3$ dissociation constants were only measured up to 6 mol kg$^{-1}$ and the effects beyond this range were extrapolated based on established relationships from experimental data (Millero et al., 1989). Building upon these extrapolations, Zhang et al. (2023) further refined the relationship between IS (< 11 mol kg$^{-1}$) with $k_1$ and $k_2$ based on laboratory-measured HMS formation rates, which already accounted for the potential uncertainties inherent in these extrapolation results for IS ranging from 6 to 11 mol kg$^{-1}$. Consequently, it was expected that the EF calculations can effectively represent the IS

effect on HMS formation rates for IS below 11 mol kg$^{-1}$. However, uncertainties in EF determinations at high IS levels (>11 mol kg$^{-1}$) may persist, potentially contributing to the notably lower $P_{HMS}$ predicted under high IS conditions, particularly on clean days with lower RH conditions. In addition, while aerosol HMS formation was typically thought to take place within the bulk aerosol water, the distinct characteristics of aerosol surface, such as a high surface-to-volume ratio, a complex and concentrated species distribution, can potentially boost the uptake of $SO_2$ and HCHO and thus facilitate the HMS formation process. For instance, the complex products involving HCHO with $H_2O$ (Ervens et al., 2003) and HMS with $SO_3^{2-}$ (Ota and Richmond, 2012) formed near or at the aerosol surface may enhance the conversion of gas-phase precursors into aqueous phase. These processes may also contribute to the disparities between the lower HMS formation rates and ambient HMS levels. Besides, above IS-dependent effects were determined for well-mixed binary aqueous system containing $Na^+$ and $Cl^-$ or $NO_3^-$, which may also introduce inaccuracies when applied to ambient aerosols. Ambient aerosols contain both water-soluble inorganic and organics together with water-insoluble organics, potentially undergoing phase separation, which may lead to two distinct phases: an inorganic-rich hydrophilic phase and an organic-rich hydrophobic phase. Given that the formation of HMS necessitates the initial hydration of $SO_2$ succeeded by the acidic dissociation of $H_2SO_3$ into $HSO_3^-$ and $SO_3^{2-}$, it is likely that the HMS formation predominantly proceeds within water-rich fraction and separated liquid phases could have differing properties (i.e., pH and ionic strength), thereby influencing the HMS formation process. Additionally, the concentrated inorganic ions in aerosol water (i.e., $SO_4^{2-}$, $NO_3^-$, $Cl^-$) can largely enhance the partitioning of gas-phase HCHO compared to pure water due to the "salting-in" effect, with effective Henry's law coefficients for HCHO being $10^2-10^4$ times higher than the classical value reported for pure water (Shen et al., 2024a; Zhao et al., 2024). A recent field-based study tried to formulate the effect of aqueous sulfate concentration ($C_{sulfate}$, in the unit of mol kg$^{-1}$ ALWC) and the effective Henry's law coefficients for HCHO. They found that while HCHO solubility did increase with rising $C_{sulfate}$ levels ranging from 1 to 9 mol kg$^{-1}$ ALWC in the winter of Beijing, hazy days experienced significantly lower HCHO solubility by 1-2 orders of magnitude due to lower $C_{sulfate}$ values as a result of rapid increase in ALWC. In this study, we also observed lower $C_{sulfate}$ levels but still estimated higher $P_{HMS}$ during haze event (**Figure S10**), further indicating the important role of ionic strength in promoting the formation and accumulation of HMS. Overall, these $P_{HMS}$ values reported here only serve as a first approximation to describe the potential daily fluctuations of HMS formation rates and further laboratory experiment conducted under more atmospherically relevant conditions (i.e., IS > 11 mol kg$^{-1}$, various aerosol compositions and aerosol phase state) is recommended to better refine our understanding of the impacts of ionic strength on particular sulfur chemistry.

More importantly, our results demonstrated that the discontinuous nature of IS-dependent enhancement on HMS formation allowed the reduced IS levels observed during humid and hazy days to effectively offset the inhibitory impact resulting from a half-unit decrease in aerosol pH, consequently resulting in enhanced HMS formation.

It also explained the negative correlation between aerosol IS and HMS level. A previous study has also noticed that incorporating the IS effect into regional models might occasionally boost HMS formation even under lower aerosol water content and elevated aerosol acidity conditions in urban Beijing, whereas the exact extent and scope of this compensatory mechanisms remain ambiguous.

**3.4 The interplay between aerosol acidity and ionic strength in HMS formation**

Sensitivity tests further revealed that a 10% decrease in aerosol acidity and ionic strength exerted notably stronger and adverse effect on HMS production rate compared to other factors including ALWC, gas precursors level and temperature (**Figure 4a**). Given the commonly observed heightened humidity conditions and resultant lower ionic strength as well as the reduction in aerosol acidity during winter haze, the interaction between aerosol acidity and ionic strength on HMS formation rates was preliminarily quantified by analyzing the $P_{HMS}$ isopleths as a function of pH and IS level under different environmental conditions (**Figure 4b-d**). Here the averaged conditions from different environments (i.e., urban haze, urban clean and marine atmospheres) as detailed in **Table S1** were utilized in the $P_{HMS}$ calculations. The green dash lines, denoted by EF=1, represented the point at which the $P_{HMS}$ value was equivalent to the $P_{HMS, dilution}$ estimated for dilute solution, mimicking conditions of cloud/fog droplets (IS < 1 mol kg$^{-1}$). As depicted by the green lines, pH-dependent promotion in $P_{HMS}$ intensified at elevated pH levels. For instance, the $P_{HMS}$ at pH 4 was approximately 30 times faster than at pH 3, and the $P_{HMS}$ at pH 6 was over 100 times faster than that at pH 5. This green dash line also exhibited a declining trend with increasing pH, suggesting that ionic strength played a more pronounced role in facilitating HMS formation within highly acidified aerosols. Furthermore, the red ridgelines at IS $\approx$ 4 mol kg$^{-1}$, which represented the IS level associated with the highest $P_{HMS}$ at a given pH, can divide the figure into 2 regions. Above the ridgelines, a pH-limited regime can be identified, where HMS production rate escalated with increasing pH but diminished with rising IS levels. Conversely, below the ridgelines, a probable co-limited regime emerged, where $P_{HMS}$ rose with both IS and pH levels, with changes amplified by variations in both factors. It is noted that consistent positioning of ridgelines across **Figure 4b-d**, suggesting that the variations in other factors such as precursors' concentrations and ALWC exhibit minimal impact on regulating HMS formation compared to aerosol pH and ionic strength. In addition, we found that the $P_{HMS}$ at the ridgelines were larger to that under 1-unit higher pH at green dash lines. These results implied that the enhancement of aerosol ionic strength level can potentially offset the inhibitory effects of more than 1-unit decrease in aqueous pH, highlighting the importance of HMS formation in aerosol water compared to dilute solutions. In this study, urban aerosols predominantly fell within the pH-limited zone given their higher IS levels resulted from the prevalence of ionic species and moderate humidity conditions (**Figure 4b** and **4d**). Therefore, although aerosol pH declined during haze days (red dots in **Figure 4b**), the enhancement from lower IS levels under higher humidity condition ultimately led to higher HMS formation rates than clean days (pink

dots in **Figure 4d**).

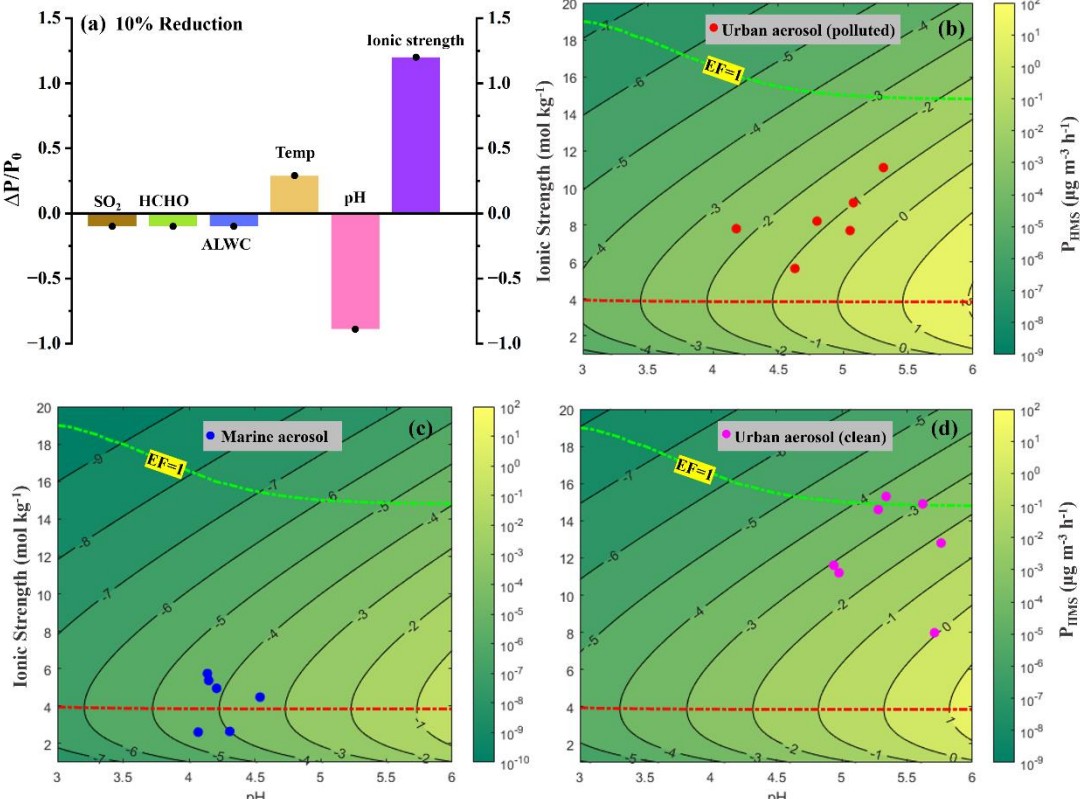

**Figure 4**. (a) The relative variation in HMS formation rate ($\Delta P/P_0$) corresponding to every 10% reduction in $SO_2$ level, HCHO level, temperature, ALWC, aerosol pH and ionic strength compared to the base scenario where $[SO_{2(g)}] = 1$ ppb, $[HCHO_{(g)}] = 1$ ppb, T=278 K, ALWC = 100 µg m$^{-3}$, aerosol pH =4, ionic strength =10 mol kg$^{-1}$. The y-axis represented the relative variation in $P_{HMS}$, calculated by the variation ($\Delta P$) relative to the original $P_{HMS}$ value ($P_0$); (b-d) The HMS formation rate ($P_{HMS}$) isopleths as a function of pH (ranging from 3 to 6) and ionic strength (ranging from 1 to 20 mol kg$^{-1}$) using the averaged atmospheric conditions from different environments as detailed in **Table S1**. The aerosol droplet radius $R_p$ was 0.15 µm (Liu et al., 2021). Every isopleth is spaced by $10^1$ µg m$^{-3}$ h$^{-1}$, with the isopleth label indicating the logarithmic value of $P_{HMS}$. The green dash line, denoted by EF=1, represented the point at which the $P_{HMS}$ value, accounting for the impact of ionic strength, equaled the $P_{HMS}$ in a dilute scenario without considering the IS effect. The red dashed line represented the ridgeline associated with the IS value that corresponds to the highest $P_{HMS}$ at a given pH.

Additionally, particulate sulfur chemistry including HMS formation has been demonstrated in coastal and marine regions (Hong et al., 2023; Zhao et al., 2024), where aerosols likely exhibited lower IS levels in comparison to continental area, owing to the moist and pristine atmospheric conditions (Song et al., 2018). In sight of this, here we also estimated the aerosol properties and the potential HMS formation rates in marine atmosphere during the cruises in YS and BS. High humidity conditions were observed, with an average value of 83%±6% (**Figure 1**). Chemical analysis of cruise samples

revealed that the ambient concentrations of $SO_{2(g)}$, HMS and sulfate averaged at $0.82\pm0.47$ ppb, $0.05\pm0.01$ µg m$^{-3}$ and $2.30\pm0.40$ µg m$^{-3}$, respectively. The estimated average characteristics of the marine aerosols included an ALWC of $14.7\pm6.40$ µg m$^{-3}$, a pH value of $4.25\pm0.15$, and an ionic strength of $4.30\pm1.25$ mol kg$^{-1}$ (**Table S1**, **Figure S11**). Notably, these marine aerosols resided in the region near the ridgeline (**Figure 4c**), where the ionic strength ranges exhibited more significant enhancements on HMS formation compared to urban aerosols. In this zone, the rate of HMS formation, when factoring in the IS effect, typically surpassed that under dilute solution where the pH was 1-unit higher (green dash line). This might imply the potential importance of HMS

10   formation in ambient aerosol even amidst the rapid acidification of sea salt particles due to IS-dependent enhancement. Using the averaged $[SO_{2(g)}]=0.82$ ppb and $[HCHO_{(g)}]=0.5$ ppb (Ervens et al., 2003; Zhao et al., 2024), the potential HMS formation rates in these marine aerosols were calculated to be $2.06 \times 10^{-5}$ µg m$^{-3}$ h$^{-1}$, a value comparable to reported cloud/fog-based HMS production rate ($1.2 \times 10^{-5}$ µg m$^{-3}$ h$^{-1}$) near Beaufort Sea (Liu et al., 2021a) with a liquid water content (LWC) 4.8 mg m$^{-3}$ and a pH of 5. Considering the minimal impact of temperature inversion during our observation in April, here we also determined the in-cloud HMS formation rate to be $4.95 \times 10^{-5}$ µg m$^{-3}$ h$^{-1}$ in our marine environment using a LWC of 4.8 mg m$^{-3}$ and a pH of 5. These results further highlighted the significance of IS-dependent enhancement in

20   HMS formation, which can render the HMS formation within aerosol water a process comparable to that observed in cloud and fog environments, particularly in humid and pristine conditions.

It was logical to see lower HMS production rates in marine aerosol compared to continental aerosols given the lower levels of gas-phase precursors, heightened aerosol acidity, and reduced aerosol water content. These factors collectively led to significantly diminished levels of aqueous-phase precursors, resulting in a lower formation rate. Therefore, we calculated the aqueous HMS conversion ratio by normalizing the HMS level against the stable aqueous concentration of $SO_2$ ($[SO_{2(aq)}]$)

30   (as shown in **Eqn. S7**), while factoring in the influences from variation in its gas-phase level, aerosol water content, and acidity, to assess the role of IS on HMS formation under various atmospheric conditions (**Figure S12**). In urban Nanjing, higher conversion ratios were observed during hazy days compared to clean days, which was likely attributed to the impact of ionic strength where lower IS levels exhibited more pronounced enhancement on HMS formation during haze events. Notably, despite lower HMS and $SO_{2(g)}$ levels observed in marine aerosols, the aqueous conversion ratio for HMS was approximately twice as high as that in urban aerosols, potentially due to the HMS-formation favored ionic strength levels (ranging from 2.0 to 6.0 mol kg$^{-1}$) in marine aerosols.

40

## 4. Conclusion

In summary, this study provided observational evidence for ambient HMS abundance in a continental city and, for the first time, in the marine atmosphere, highlighting the ubiquity of HMS in mildly polluted and clean marine environments. Relatively lower HMS levels were observed in this work compared to those reported in Northern China. This divergence can be largely attributed to the likely minor contribution of HMS formed in cloud/fog processes. Despite this, a notable increase in HMS concentration was recorded during a 7-day haze episode in urban Nanjing. Correlation analysis highlighted that variations in HMS levels were predominantly influenced by HCHO levels, relative humidity, and aerosol properties. The significant correlation between HMS levels and higher RH has previously been associated with elevated aerosol liquid water content, which allowing for more efficient dissolution of gas-phase precursors. In urban Nanjing, we also observed increased RH and HMS level during haze event (**Table S1**). Besides, ionic strength also showed a high dependence on ambient humidity (R=-0.89) even under high $PM_{2.5}$ scenarios, with 10% increase in RH level leading to nearly 2-units reduction in aerosol ionic strength.

Notably, even with a 4-fold increase in ALWC coupled with a 2-fold increase in HCHO levels, these factors alone cannot completely counterbalance the potential 10-fold reduction in HMS formation rates resulting from the decreased aerosol pH, which failed to explain the heightened HMS formation during pollution days. However, lower aerosol ionic strength on humid and polluted days was found to exhibit more pronounced enhancement in HMS formation, ultimately leading to a nearly 10-fold rise in $P_{HMS}$ during haze episode compared to the dry and clean days. This can be explained by the fact that the impact of ionic strength on HMS formation rate exhibited non-uniform responses to variations in IS levels as the rising IS can boost the reaction rates of **R1** and **R2** while also impeding the solubility $SO_2$ and affecting the dissociation constants of $H_2SO_3$. Specifically, the IS-dependent enhancement on HMS formation rate first increased with elevating IS levels, peaking at IS of around 4 mol kg$^{-1}$, before declining. Furthermore, the dynamic interplay between aerosol ionic strength and acidity were quantified through the delineation of $P_{HMS}$ isopleths. The result indicated the peak enhancements exerted by IS can substantially promote the HMS formation, yielding a $P_{HMS}$ value comparable to that observed under diluted cloud/fog solution with a 2-units increase in aqueous pH. More importantly, these results suggested higher relative humidity (RH) levels observed during hazy days can not only provide a greater amount of aerosol liquid water content but also dilute the urban aerosol solution towards an IS level that exhibited peak enhancement on HMS formation. Therefore, moderate IS levels presented in humid environments and their enhancing effects may play a crucial role in HMS formation on urban pollution days, despite the common decrease in aerosol pH ($\Delta$pH<-1) observed during haze events (Sha et al., 2019; Ruan et al., 2022; Fu et al., 2024; Zhang et al., 2024). Additionally, lower IS levels (2.0-6.0 mol kg$^{-1}$), prevalent in marine aerosols, can significantly promote HMS formation within aerosol water, making this process comparable with that in cloud/fog droplets in marine

atmosphere. Although the HMS formation rates were 1-2 orders of magnitude lower in marine atmosphere compared to urban atmosphere, aqueous conversion ratio for HMS from aqueous $SO_{2(aq)}$, when normalizing the ambient HMS level to gas-phase concentration of $SO_2$, aerosol liquid water content and acidity, was nearly 2 times higher in marine aerosols than in urban aerosols, likely attributed to their conducive ionic strength level to HMS formation.

In addition, laboratory studies have reported that although HMS is resistant to oxidation by $H_2O_2$ or $O_3$, it can be oxidized by $^{\bullet}OH$ through aqueous or heterogenous reactions, eventually leading to the formation of $SO_4^{2-}$ after a series of radical chain reactions (Seinfeld et al., 2016; Lai et al., 2023). Model simulations also indicate that this reaction can lead to a reduction of atmospheric concentrations of HMS by 10-20% in winter and 40-60% in summer on a global scale (Song et al., 2021). More importantly, model studies underscore the potentially critical role of HMS as an atmospheric sulfur reservoir in sulfate formation (Dovrou et al., 2022; Zhao et al., 2024). This significance arises from the fact that the $HCHO_{(aq)}$ can be released during the oxidation of HMS, which can react with $SO_2$ to regenerate HMS and be subsequently released back into the aqueous phase during further HMS oxidation by $^{\bullet}OH$. Under such circumstance, HMS can be considered as atmospheric sulfur reservoir with $HCHO_{(aq)}$ acting as the catalyst that facilitates the continuous conversion from gas-phase $SO_2$ to $SO_4^{2-}$. A recent modeling work indicated that this $HCHO_{(aq)}$ catalysis process could contribute up to 20-30% of particulate sulfur formation under coastal and marine conditions (Zhao et al., 2024), despite that uncertainties may persist due to the lack of laboratory-based kinetic parameters for HCHO-related pathways and the oversight of IS effects in HMS and sulfate formation. Furthermore, considering the moderate ionic strength level in marine aerosol and the fact that photodecomposition of Enteromorpha can release substantial amounts of formaldehyde to marine atmosphere (Shen et al., 2024b), it is conceivable that HMS may play a more critical role in particulate sulfur chemistry under marine conditions. Collectively, this study provided valuable information for the prevalence of HMS and the validation the model-derived outcomes concerning HMS quantification. The work primarily concentrated on particulate HMS formation in aerosol liquid water, highlighting the role of moderate-level ionic strength in atmospheric HMS formation, advocating for their integration into global or regional models to better represent the particulate sulfur chemistry, especially in humid environments. Nevertheless, it is noted that in-cloud HMS chemistry may also contribute to the particulate HMS levels where vertical and high-altitude observations are required to fully understand its significance, thus warranting further investigation.

**Data availability.** Data are available upon request from the corresponding author.

**Supplement.** Additional information on Collection and Chemical Analysis of Ambient Samples (Section S1); HCHO Concentration Retrieval from MAX-DOAS (Section S2); Backward Trajectories Analysis (Section S3); Aerosol Properties Estimations (Section S4); Steady-state HMS Formation Rate ($P_{HMS}$) Calculation (Section S5); Aqueous HMS Conversion ratio Calculation (Section S6).

**Author Contribution.** RX and YZ designed this study. RX, and SB coordinated field sampling and data accusation. FX assisted in the data visualization. RX performed the data analysis and wrote the original paper. YCL and YZ reviewed the paper and provided comments and suggestions.

**Competing interests.** The contact author has declared that neither they nor their co-authors have any competing interests.

**Acknowledgment**. We would like to thank Prof. Cheng Liu's group at University of Science and Technology of China for providing the hyperspectral vertical profile retrieval method and the valuable HCHO vertical profiles during our sampling period in Nanjing. These profiles support the calculation of urban HMS formation rates in our study. We also thank Zeliang Huang at Nanjing University of Information Science and Technology for assisting the marine samples collection.

**Financial support**. This work was funded by the National Natural Science Foundation of China (project number: 42303083), and the Key Program of the National Natural Science Foundation of China (project number: 42192512). Marine data and samples were collected onboard of R/V "Lanhai101" implementing the open research cruise NORC2024-01 supported by NSFC Shiptime Sharing Project (project number: 42349901).

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
