# Peer review of "Hydroxymethanesulfonate (HMS) Formation under Urban and Marine Atmosphere: role of aerosol ionic strength"

_EGUsphere, 2025_

## Author Response (AR1)

*This paper investigates HMS formation as a function of particle pH and ionic strength (IS) based on two data sets: an urban site and a marine site. Modeling shows interesting interaction between IS and pH on HMS production rates. The concentrations of HMS are very low and a very small fraction of PM2.5 mass in both locations so it is unclear why the authors are attempting this analysis with this data – the contribution of HMS seems of little importance in these data. I guess the belief is the findings here have broad applicability. This could be made clearer. The analysis assume the HMS was formed under the conditions in which the measurements were made, otherwise the analysis falls apart. Other than low wind speeds, there is little proof offered to support this. Furthermore, how do the authors know that the formation didn't occur in clouds not the aerosol, which are much more conducive to HMS formation. More evidence of formation in ALWC would be helpful. Aerosol pH is a critical parameter, but no evidence is provided to suggest it is reasonably predicted and assumptions are made due to lack of ammonia data. Extensive modeling analysis is performed, and the results are interesting, but little evidence is given that the data strongly support the model predictions; the main point seems to be that the model and data agree at the level of general trends in pH and IS. A more in-depth comparison might be helpful. A more direct comparison between model and measurements would be useful. Finally, given all the caveats stated in this manuscript relating to uncertainties in rate constants, phase separations, and possible surface vs bulk particle reactions, how does one interpret the modeling results as being reliable? This is why more direct comparisons would be useful. Overall, the modeling results are very interesting but there could be improvements to the manuscript. More details are given below.*

**General Response:**

We thank the reviewer for his/her insightful and constructive comments, which have greatly assisted us in enhancing this paper.

- First of all, we concur with the reviewer that cloud water could serve as a significant HMS formation medium and more declaration should be given to elucidate the contribution of in-cloud HMS formation to our observations. In the wintertime of Nanjing, characterized by weak vertical exchange due to temperature inversions, low wind speeds, and reduced cloud water content, it is plausible that the impact of in-cloud HMS formation on the observed levels of particulate HMS may be insignificant (please see our response to **Major Comment #2** for further details). We acknowledge that this study observed lower levels of HMS compared to polluted areas such as Northern China, possibly due to the minor contribution from in-cloud formation. However, our observations in Nanjing could offer

valuable insights into the aerosol chemistry related to HMS formation. Furthermore, despite the lower HMS levels, this study has provided supplementary information on ambient HMS abundance in less polluted and even marine pristine environments, enhancing our understanding of the prevalence of particulate sulfur species.

- Regarding the aerosol pH estimations, we acknowledge the need for clearer descriptions. We have monitored the gas-phase $NH_3$ in Nanjing using a Monitor for AeRosols and GAses (MARGA; Metrohm Ltd., Switzerland). And we adopted a particle to particle + gas partitioning fraction of $NH_4^+$ ($\varepsilon$) of 0.5 to estimate the marine aerosol pH, drawing from prior observations conducted in the Bohai Sea (refer to our reply to **Specific Comment #7**).

- Moreover, we have expanded on the discussion comparing our model results with ambient measurements, as outlined in our responses to **Specific Comment #15**.

Detailed point-to-point responses to the referee's comments are presented below in regular font, while the referee's comments are highlighted in italics.

**Major Comment #1:**

*HMS was measured by ion chromatography. Regarding the detailed discussion in the supplemental material page 10 and Fig S10. Work by (Wei et al., 2020; Lai et al., 2023) suggest that HMS is actually detected as sulfite since it is converted to other or free S(IV) species in the IC column due to the eluent. To address this, a method of adding H2O2 to a filter sample extract to convert free bisulfite and sulfite to sulfate. This is found to have a minor effect since the untreated and treated samples give similar HMS concentrations. However, a similar treatment in Fairbanks shows that this sample treatment results in significant changes in HMS concentrations and that this is likely associated with other aldehyde-S(IV) adducts See Dingilian et al, ES&T Air 2024 https://doi.org/10.1021/acsestair.4c00012 There is evidence for these other aldehyde-S(IV) adducts in NCP hazes from single particle analysis, see https://doi.org/10.5194/acp-20-5887-2020 and https://doi.org/10.1016/j.envpol.2022.120846, although they may not be a large fraction of S(IV) measured. This may not affect the results of this study, but a brief discussion of this in the main text would be useful – the whole paper depends on the assumption that HMS is what is being measured by the IC.*

**Response:** Thanks for pointing this out. We agree with reviewer that free S(IV) or other S(IV) species such as aldehyde-S(IV) adducts may be misidentified as HMS during the IC analysis. In our chemical analysis process, an aliquot of these sample extracts was directly injected into IC and the second aliquot was treated with 0.1 mL of 3% $H_2O_2$ before injection to address the potential inaccuracies arising from free S(IV) in the quantification of HMS, as detailed in Supporting Information (**Section S1**). This method can be extended to other S(IV) species like aldehyde-S(IV) adducts, given $H_2O_2$ can convert both free S(IV) (e.g., sulfite and bisulfite) and other S(IV) species such as aldehyde-S(IV) adducts to sulfate (Dingilian et al., 2024). **Notably, the deviations in**

**HMS quantification resulted from free S(IV) and other S(IV) species were found to be less than 5%** (**Figure S13c**). The following discussions were added in the Manuscript and Supporting Information (highlighted in blue).

**Revision in Manuscript:**

**Page 5, Line 3:** "Given that free S(IV) or other S(IV) species such as aldehyde-S(IV) adducts may be misidentified as HMS during the IC analysis (Ma et al., 2020; Dingilian et al., 2024), hydrogen peroxide ($H_2O_2$) was employed in analysis process to specifically isolate HMS (Dingilian et al., 2024). The results suggested the influence of free S(IV) or other S(IV) species on HMS measurement was negligible, as elaborated in the Supplementary Information (**Section S1**)"

**Revision in Supporting Information:**

**Page 13, Line 22:** "It is noted that HMS was detected in the form of sulfite in IC analysis as it can readily decompose into sulfite upon mixing with the alkaline eluent (Wei et al., 2020; Lai et al., 2023). Therefore, free S(IV) (i.e., sulfite and bisulfite) or other S(IV) species such as aldehyde-S(IV) adducts may be misidentified as HMS during the IC analysis (Ma et al., 2020; Dingilian et al., 2024). Thus, to determine the potential error in HMS quantification raising from free S(IV) and aldehyde-S(IV) adducts, 0.1 mL of 3% $H_2O_2$ was added into the second aliquot of the closed sample extracts, which converted other S(IV) species to sulfate over a reaction period of 3 hours (Dingilian et al., 2024). Before injected into IC, 0.1 mL volume of $H_2O_2$ catalase solution (>200.000 unit/g, Aladdin) was added to destroy any excess $H_2O_2$. As shown in **Figure S13c**, here we tested 3 samples and found that the deviations in HMS quantification resulted from other S(IV) species were less than 5%, consistent with previous findings (Wei et al., 2020). "

**Major Comment #2:**

*Why does one assume that the conditions in which the aerosol is being measured represents the conditions under which HMS was formed under. Low wind speed seems the only basis. This argument depends on the estimated lifetime of HMS, ie that it has a short lifespan. If the HMS (and possibly sulfate) was formed in clouds all of what is presented in this manuscript is not correct. This is especially true for the marine environment where cloud processing is frequent and expected. It is noted in the manuscript (pg 4 line 3 and on) that only 33% of HMS is thought to be formed in aerosol particles over the NCP, with 66% in cloud and fogs, so why all in aerosols in this study? This is critical and should be given more attention.*

**Response:** Thanks for raising this critical point. We agree with the reviewer that cloud could serve as a significant medium for HMS formation due to the large amount of cloud water content (CWC). It is noted that the one-third contribution of HMS formed within aerosols (HMSp) to total HMS levels was a general value during their observation period from November 26[th] to December 16[th], 2015, Beijing (Wang et al.,

2024). However, near-surface HMSp was found to be comparable to those formed in cloud/fog water (HMSc) during their Episode 1. Moreover, under certain days (e.g., November 28th to 29th), HMSp levels could exceed HMSc level, which was attributed to that the enhancing effect of IS can counterbalance the constraints imposed by lower aerosol water content and pH values compared to cloud water. However, the exact extent and scope of this compensatory mechanisms remain ambiguous. In addition, another work has reported the detection of HMS in aerosols regardless of the presence of cloud/fog in the winter of Beijing, and HMS concentrations were found to show a good correlation (R= 0.92, P< 0.01) with aerosol liquid water content (ALWC) (Ma et al., 2020). Therefore, they proposed that aerosol water can serve as a medium for HMS formation despite the fact that HMS levels were significantly lower ($< 1\ \mu g\ m^{-3}$) in the absence of cloud/fog and peak HMS level of 18.5 $\mu g\ m^{-3}$ was observed during fog processes. Furthermore, previous study has emphasized that although in-cloud formation processes can be much faster compared to aerosol water given its large water content, their contribution to the near-surface aerosol composition can be negligible during winter haze in Northern China due to the accompanied temperature inversions and high atmospheric stability so that the weak vertical exchange impeded the transportation of gas precursors emitted near surface to high altitudes and the chemicals produced in the high-altitude cloud could not be easily transported to the ground (Wang et al., 2022b).

In our current work, no prolonged fog events (RH > 90% and Visibility < 1 km) lasting over two hours were observed as shown in **Figure S2e**, which was attached below for the reviewer's reference. Here, we utilized the vertical temperature profile obtained from http://weather.uwyo.edu/upperair/sounding.html to identify temperature inversion in the atmospheric boundary layer in the winter of Nanjing. As shown in **Figure S3**, a predominant occurrence of temperature inversions was noted across our observation days, characterized by inversion layers extending from the surface to altitudes of 2000 m, particularly evident on hazy days. Besides, days without temperature inversions (i.e., from December 19th to December 23th, 2023) were associated with negligible cloud water content (**Figure S2f**), which was obtained from MERRA-2 (Modern-Era Retrospective analysis for Research and Applications, Version 2) (Gelaro et al., 2017). In line with Ma et al.'s work (2020), we also detected HMS within PM$_{2.5}$ samples during 7 of non-cloud days (**Figure S2f**). Besides, our data indicated no discernible correlation between CWC and HMS levels whereas HMS levels displayed a notably positive correlation with ALWC (R=0.67, P<0.01). **While we recognize that the HMS formation in cloud layers characterized by persistent dense clouds and extended durations may carry great contributions, in the context of this study, the presence of temperature inversions, low wind speeds, and diminished cloud water content during the winter season in Nanjing suggests that the impact of in-cloud HMS formation on the observed levels of particulate HMS may be insignificant**. **Section 3.2** has been extensively revised to provide a more detailed explanation of the role of in-cloud formation in observed particulate HMS levels during our winter observations in Nanjing. We have also made edits in the **Abstract**.

**Revision in Abstract:**

**Page 2, Line 1:** "Hydroxymethanesulfonate (HMS) has emerged as a critical organosulfur species in ambient aerosols, yet the impact of aerosol properties, particularly ionic strength (IS), on the formation of HMS remains uncertain. Here, HMS levels in wintertime of urban Nanjing, China were quantified at 0.30±0.10 μg m$^{-3}$, where the contribution of in-cloud formation likely carry minor significance due to the barrier resulted from stable stratification. "

**Revision in Manuscript:**

**Page 8, Line 1: "**

**3.2 Elevated HMS level during a haze pollution period in urban Nanjing**

In addition, a seven-day haze pollution event (from December 27$^{th}$, 2023 to January 2$^{nd}$, 2024) was observed in urban Nanjing, where a notably elevated levels of HMS and sulfate were observed (p<0.05). Throughout the haze event (PM$_{2.5}$ = 114.3±18.0 μg m$^{-3}$), the average concentrations of HMS and sulfate were 0.36±0.09 and 11.4±4.0 μg m$^{-3}$, respectively, with HMS/sulfate molar ratio of 3.5±0.7% (**Table S1**). It is noted that even under hazy days, the HMS level were significantly lower compared to those reported in Northern China during severe winter haze episode (averaged at 4-7 μg m$^{-3}$) (Ma et al., 2020; Liu et al., 2021; Chen et al., 2022; Wang et al., 2024). Such divergence can be largely attributed to the contribution of HMS formed in cloud/fog processes, as evidenced by a previous study reporting comparable levels of HMS (< 1 μg m$^{-3}$) in Beijing to this work in the absence of cloud/fog, while peak HMS levels of 18.5 μg m$^{-3}$ was observed during fog processes (Ma et al., 2020). During our sampling period in Nanjing, there were no prolonged fog events (RH>90% and Visibility<1 km) lasting over two hours (**Figure S2e**). Additionally, previous study has emphasized that although in-cloud formation processes can be much faster compared to aerosol water given its large water content, their contribution to the near-surface aerosol composition can be negligible during hazy days in Northern China due to the accompanied temperature inversions and high atmospheric stability so that the weak vertical exchange impeded the transportation of gas precursors emitted near surface to high altitudes and the chemicals produced in the high-altitude cloud could not be easily transported to the ground (Wang et al., 2022). Here, we utilized the vertical temperature profiles retrieved from http://weather.uwyo.edu/upperair/sounding.html to identify temperature inversion within atmospheric boundary layer in the winter of Nanjing. As shown in **Figure S3**, a predominant occurrence of temperature inversions was noted across our observation days, characterized by inversion layers extending from the surface to altitudes of 2000 m, particularly evident on hazy days. Furthermore, days lacking temperature inversions (from December 19$^{th}$ to December 23$^{rd}$, 2023) were associated with negligible cloud water content (**Figure S2f**), which were obtained from MERRA-2 (Modern-Era Retrospective analysis for Research and Applications, Version 2) (Gelaro et al., 2017). In line with Ma et al.'s work (2020), we also detected HMS within PM$_{2.5}$ samples during 7 of non-cloud days. Therefore, given the protective effects of temperature inversion and low wind speeds (**Figure S2a**) as well as reduced

cloud water content prevalent during our observations, the contribution of in-cloud/fog HMS formation to observed particulate HMS levels was likely insignificant and the notably lower HMS level further suggest that the aerosol bulk water might serve as a predominant medium for HMS formation during our observations.

Other factors such as the level of precursors and atmospheric oxidants may also contribute to this divergence. For instance, during the winter haze in Beijing, the average level of HMS, sulfate were reported to be 4 µg m$^{-3}$ and 45 µg m$^{-3}$, respectively, with HMS/sulfate up to 10%, and the gaseous $SO_2$ and HCHO concentrations were recorded at 12 and 20 ppb, respectively (Ma et al., 2020). This study observed significantly lower levels of $SO_2$ (2.2±0.7 ppb) and HCHO (5.4±1.1 ppb) during the haze event. Thus, the lower HMS concentrations were expected. On the other hand, the reduced HMS/sulfate ratio could potentially be elucidated by the higher $O_3$ level in this work (43.3±4.7 ppb) compared to Beijing (5 ppb) as low atmospheric oxidation level could encourage more $SO_2$ to participate in the formation of HMS rather than sulfate (Song et al., 2019; Ma et al., 2020; Campbell et al., 2022)."

**Revision in Supporting Information: "**

[Figure]

**Figure S2.** Ancillary atmospheric measurements in urban Nanjing including fog event criteria based on hourly relative humidity (RH>90%) and Visibility (<1 km) data (e); Time series of the average cloud water content below the planetary boundary layer height over our observation sites (f).

[Figure]

**Figure S3**. Observed vertical temperature profile at 00 UTC during our observation in Nanjing. The dates of hazy days were marked by red."

As for marine environment, we attempt to determine the HMS formation rate in cloud water ($P_{HMS, c}$) and aerosol water ($P_{HMS,a}$) considering the minimal impact of temperature inversion during our observation in April. In the manuscript, we have compared the $P_{HMS, a}$ ($2.06 \times 10^{-5}$ μg m$^{-3}$ h$^{-1}$) in our study with $P_{HMS, c}$ ($1.2 \times 10^{-5}$ μg m$^{-3}$ h$^{-1}$) estimated for observations near Beaufort Sea with a CWC of 4.8 mg m$^{-3}$ and a pH of 5 (Liu et al., 2021). Using these same CWC and pH, we calculated the averaged $P_{HMS,c}$ of $4.95 \times 10^{-5}$ μg m$^{-3}$ h$^{-1}$ in our marine environment, which was comparable to the $P_{HMS, a}$. The calculation results and input parameters were summarized in **Table R1**. This comparable $P_{HMS,a}$ and $P_{HMS,c}$ can be explained by despite that the CWC being ~ 300 times higher than ALWC and aerosol pH being lower, the higher ionic strength level in marine aerosol water can significantly promote HMS formation process, rendering the HMS formation within aerosol water a process comparable to that observed in cloud and fog environments. Altogether, although these formation rates serve as initial approximations, **these results may indicate the significant contribution of both aerosol and cloud water in the formation of HMS in marine atmosphere**. Following discussion has been included in **Section 3.4**.

**Revision in Manuscript:**

**Page 17, Line 10:** "Using the averaged $[SO_{2(g)}]=0.82$ ppb and $[HCHO_{(g)}]=0.5$ ppb (Ervens et al., 2003; Zhao et al., 2024), the potential HMS formation rates in these marine aerosols were calculated to be $2.06 \times 10^{-5}$ µg m$^{-3}$ h$^{-1}$, a value comparable to reported cloud/fog-based HMS production rate ($1.2 \times 10^{-5}$ µg m$^{-3}$ h$^{-1}$) near Beaufort Sea (Liu et al., 2021) with a liquid water content (LWC) 4.8 mg m$^{-3}$ and a pH of 5. Considering the minimal impact of temperature inversion during our observation in April, here we also determined the in-cloud HMS formation rate to be $4.95 \times 10^{-5}$ µg m$^{-3}$ h$^{-1}$ in our marine environment using a LWC of 4.8 mg m$^{-3}$ and a pH of 5. These results further highlighted the significance of IS-dependent enhancement in HMS formation, which can render the HMS formation within aerosol water a process comparable to that observed in cloud and fog environments, particularly in humid and pristine conditions."

Besides, for the reviewer's reference, here we also attempt to compared the HMS formation rate in cloud water ($P_{HMS, c}$) and aerosol water ($P_{HMS, a}$) in urban Nanjing although the transportation of in-cloud HMS formation can be largely impeded. Global models (Song et al., 2019) have estimated the cloud pH in the winter of Nanjing aera to be around 4-5, consistent with winter observations in China (4.1±0.6) (Shah et al., 2020). In our analysis, we utilized the upper value of cloud pH of 5 to estimate $P_{HMS, c}$ under pre-haze and haze conditions, comparing it with $P_{HMS, a}$. Despite the tendency for cloud pH to decrease during haze pollution (Li et al., 2017), it was observed that $P_{HMS, c}$ was five times lower than $P_{HMS, a}$ on hazy days. Nevertheless, it is worth noting that the $P_{HMS, a}$ value of $1.06\pm0.64 \times 10^{-2}$ µg m$^{-3}$ h$^{-1}$ can effectively reflect the HMS levels observed in ambient aerosol, furthering indicate the predominate role of aerosol water for HMS formation during temperature inversion. Besides, the $P_{HMS, c}$ were found to be comparable to $P_{HMS, a}$ on clean days (**Table R1**). Considering possible overestimations in $P_{HMS, c}$ due to reduced precursor concentrations and hindered transport to ground level resulting from weak vertical exchange, these results suggest the limited impact of in-cloud HMS formation on detected HMS levels. We acknowledge that uncertainties may exist in the calculation of HMS formation rate due to potential variability associated with cloud pH, thus above information was not included in the Manuscript.

**Table R1**. Summary of $P_{HMS, a}$ and $P_{HMS, a}$ as well as input parameters.

| | Marine | | Urban | | | |
|---|---|---|---|---|---|---|
| | **Cloud** | **Aerosol** | **Cloud** | | **Aerosol** | |
| | | | Pre-haze | Haze | Pre-haze | Haze |
| T (K) | 283±1 | 283±1 | 272±2 | 277±4 | 272±2 | 277±4 |
| SO$_2$ (ppb) | 0.82±0.42[a] | 0.82±0.42 | 1.99±0.45 | 2.21±0.69 | 1.99±0.45 | 2.21±0.69 |
| HCHO (ppb) | 0.5[a] | 0.5 | 1.91±1.05 | 5.38±1.08 | 1.91±1.05 | 5.38±1.08 |
| LWC (µg m$^{-3}$) | 4.8[b] mg m$^{-3}$ | 14.72 | 1.5±2.2 mg m$^{-3}$ | 5.12±5.45 mg m$^{-3}$ | 22.69±20.65 | 80.45±38.40 |

| pH | 4.8 [b] | 4.24±0.15 | 5 [d] | 5 | 5.25±0.42 | 4.76±0.46 |
|---|---|---|---|---|---|---|
| Ionic strength (mol kg$^{-1}$) | 10$^{-4}$ [c] | 4.30±1.24 | 10$^{-4}$ | 10$^{-4}$ | 12.32±3.19 | 8.85±1.30 |
| $P_{HMS}$ (× 10$^{-4}$ μg m$^{-3}$ h$^{-1}$) | 0.51 | 0.21 | 2.56±4.68 | 20.4±23.3 | 6.40±6.96 | 106±64.6 |

[a] The levels of SO$_2$ and HCHO over marine environments were adopted from previous studies (Ervens et al., 2003; Zhao et al., 2024).
[b] The LWC and pH for cloud/fog droplets came from previous coastal study near Beaufort Sea (Liu et al., 2021).
[c] This typical ionic strength value of cloud droplet was given by Herrmann et al. (2015).
[d] This cloud PH was the upper value of global estimation on Nanjing aera (Song et al., 2019).

**Major Comment #3:**

*In this study HMS is a very small fraction of sulfate both in the urban and marine locations. Why is it so small compared to other studies in the NCP. Seems like it is not really that important since it is a very small mass component of PM$_{2.5}$. This is discussed on page 8, but one could add a justification that the analysis of IS effects can still be applied even if the concentrations of HMS are low. Is this why HMS is not shown on Fig 1d that shows the chemical speciation of PM$_{2.5}$ (it is shown in Fig 1h, which is odd). Maybe the low concentrations reflect that it was not formed in cloud/fog water.*

**Response:** Thanks for the comment. We acknowledge that the HMS levels observed in urban Nanjing are notably lower than that in NCP region. We agree with the reviewer that such divergence can be largely attributed to the limited contribution of HMS formed in cloud/fog processes as a previous study has reported comparable levels of HMS (< 1 μg m$^{-3}$) in Beijing to this work in the absence of cloud/fog, while peak HMS levels of 18.5 μg m$^{-3}$ was observed during fog processes (Ma et al., 2020). The detailed explanation on the limited in-cloud formation contribution can be referred to our reply to **Major Comment #2**. In addition, other factors such as the level of precursors and atmospheric oxidants may also contribute to this divergence. For instance, during the winter haze in Beijing, the average level of HMS, sulfate were reported to be 4 μg m$^{-3}$ and 45 μg m$^{-3}$, respectively, with HMS/sulfate of 10%, and the gaseous SO$_2$ and HCHO concentrations were recorded at 12 and 20 ppb, respectively (Ma et al., 2020). This study observed significantly lower levels of SO$_2$ (2.2±0.7 ppb) and HCHO (5.4±1.1 ppb) during the haze event. On the other hand, the reduced HMS/sulfate ratio could potentially be elucidated by the higher O$_3$ level in this work (43.3±4.7 ppb) compared to Beijing (5 ppb) as low atmospheric oxidation level could encourage more SO$_2$ to participate in the formation of HMS rather than sulfate (Song et al., 2019; Ma et al., 2020; Campbell et al., 2022). Above information has been included in the Manuscript. And we have added HMS level into the **Figure 1** as attached below.

Regarding the importance of ionic strength in this lower HMS formation observed in our study, we have compared the HMS formation rates with and without the consideration of IS in urban Nanjing (**Figure S9**, attached below). And despite the low reactivity of HMS after its formation, other atmospheric process such as physical

transport and deposition could also impact its ambient level. Therefore, we has utilized the HMS to carbon dioxide (CO) ratio to better represent the secondary formation of ambient HMS as CO was usually considered to be an inert chemical species during rapid haze formation, with its variation often interpreted as indicative of the accumulatio of primary pollutants in the shallower boundary layer (Williams et al., 2016). During our study period, enhanced HMS formation was noted during hazy days as evident by the higher HMS/CO ratio (0.51±0.11) and elevated HMS formation rates were found during haze event with averaged $P_{HMS}$ of $5.8\pm5.9\times10^{-3}$ µg m$^{-3}$ h$^{-1}$ compared to that during the clean periods ($4.5\pm8.4\times10^{-4}$ µg m$^{-3}$ h$^{-1}$) (**Figure 2f**, attached below). These $P_{HMS}$ estimations exhibited a good correlation with HMS/CO (R=0.57) (**Figure S9a**). However, when the ionic strength effect in aerosol water was not considered, daily $P_{HMS}$ was generally calculated to be one to two orders of magnitude lower (**Figure S9b**). This was due to that even with a 4-fold increase in ALWC coupled with a 2-fold increase in HCHO levels, these factors cannot completely counterbalance the potential 10-fold reduction in HMS formation rates resulting from the decreased aerosol pH, leading to slower HMS formation rates ($1.2\pm1.7\times10^{-4}$ µg m$^{-3}$ h$^{-1}$) during hazy days compared to clean days ($3.3\pm5.3\times10^{-4}$ µg m$^{-3}$ h$^{-1}$) (**Figure S9c**), failing to explain the higher HMS/CO ratio. As for marine environments, our calculations in reply to **Major Comment #2** has suggested that although the CWC being ~ 300 times higher than ALWC and aerosol pH being lower, the higher ionic strength level in marine aerosol water can significantly promote HMS formation process, rendering the HMS formation rate within aerosol water ($2.06 \times 10^{-5}$ µg m$^{-3}$ h$^{-1}$) comparable to that observed in cloud and fog environments ($4.95 \times 10^{-5}$ µg m$^{-3}$ h$^{-1}$). Taken together, these results emphasize the significance of aerosol ionic strength in HMS formation during our observations.

[Figure]

**Figure 1.** Left panel (Urban Observations): Time series of atmospheric measurements

in urban Nanjing, including temperature and relative humidity (a); $SO_2$ and HCHO levels (b); HMS and sulfate concentration (c); $PM_{2.5}$ composition (d). The gray shade area highlighted a 7-days' haze episode (December 27th, 2023 to January 2nd, 2024) with $PM_{2.5}$ mass concentration exceeding 75 $\mu g\ m^{-3}$. Other ancillary measurements in urban atmosphere including wind speed and wind direction, $PM_{2.5}$ mass concentration, $O_3$, $NH_3$ level, fog occurrence and cloud water content can be found in **Figure S2**. Right panel (Marine Observations): Atmospheric measurements during marine cruise, including temperature and relative humidity (e); $SO_2$ (f), HMS and sulfate level where the biogenic nss-sulfate (bio-nss-sulfate) was calculated based on previously measured bio-nss-sulfate/nss-sulfate ratio of 0.14 (Yang et al., 2015) (g); $PM_{2.5}$ composition (h).

[Figure]

**Figure S9.** (a) The comparison between estimated HMS formation rate ($P_{HMS}$) with HMS/CO ratio in urban Nanjing; (b) The comparison between $P_{HMS}$ with and without considering the IS effect in urban Nanjing; (c) the averaged $P_{HMS}$ before, within and after haze event, with and without consideration of ionic strength. The formation rate estimations on December 19th, 2023 and January 3rd, 2024 are not feasible due to the absence of HCHO level and aerosol properties, respectively; (d) The relative variation in HMS formation rate during hazy days ($\Delta P/P_{HMS,\ clean}$) corresponding to changes in $SO_2$ level, HCHO level, temperature, ALWC, pH and ionic strength compared to clean days.

[Figure]

**Figure 2f.** Comparison between HMS/CO ratio and HMS formation rates ($P_{HMS}$).

**Major Comment #4:**

*From the chemical speciation it seems the most important $PM_{2.5}$ component by far is ammonium nitrate. Maybe that should be pointed out. The interesting thing about the prevalence of nitrate is that it suggests a certain pH range (roughly 3 or higher) https://doi.org/10.5194/acp-18-12241-2018. Might want to note that this is roughly consistent with the calculated pH. Furthermore, it would be worth noting what chemical species drove ALWC, seem to be nitrate (Section 3.3).*

**Response:** Thanks for the suggestion and bringing up this paper. As suggested by Guo et al. (2019), we calculated the $\varepsilon(NO_3^-)$ during our observations, which is defined as the fraction of $NO_3^-$ in the particle phase relative to total nitrate ($HNO_3$+ $NO_3^-$). According to the relationship between $\varepsilon(NO_3^-)$ and aerosol pH proposed by Guo et al. (2019), the $\varepsilon(NO_3^-)$ values in this work (>95%) indicates a prevailing high pH environment approximating 5. These results showed a good consistency with the estimated aerosol pH range in our work. In addition, we also utilized a random forest (RF) model with Shapley additive explanations (SHAP) to identify crucial factors influencing aerosol pH, as illustrated in previous studies (Li et al., 2024; Zhang et al., 2025). Details of the parameters and specifications of this method were added in the Supporting information (as shown in our revision below). Briefly, the model input includes T, RH, Weed Speed (WS), boundary layer height (BLH), $PM_{2.5}$, $SO_4^{2-}$, $NO_3^-$, $NH_4^+$, $SO_2$, $O_3$, $NH_3$ and pH. It is noted that previous study has point out that RH, as a more general environmental parameter, is more suitable as an input parameter than ALWC for assessing aerosol pH, given the quantitative relationship between pH and ALWC (Zhang et al., 2025). Consequently, ALWC was excluded from the input factors. As shown in **Figure S16a**, the aerosol pH predicted by RF model closely matches our calculation from thermodynamic models, with $R^2$ of 0.96. And RH has the highest contribution (**Figure S16b**) and largest SHAP value (**Figure S16c**) on aerosol pH. Besides, $SO_4^{2-}$ rank as the second most important factors, surpassing the contribution of $NO_3^-$. This result was in consistent with a previous work in NCP region (Ding et al.,

2019), which declared that $SO_4^{2-}$ had a greater effect than $NO_3^-$ on $PM_{2.5}$ pH as $SO_4^{2-}$ can lead to a higher concentration of $H_{air}^+$ than $NO_3^-$ due to its low volatility and strong dissociation. We have added above information into the Manuscript and Supporting Information (**Section S4**).

**Revision in Manuscript:**

**Page 11, Line 23:** "The RH conditions and sulfate levels were identified as pivotal factors influencing pH fluctuations in this study based on a Random Forest model combined with Shapley additive explanations (SHAP) analysis (Zhang et al., 2025) (as detailed in **Section S4**)."

**Revision in Supporting Information:**

**Page 17, Line 32:** "In addition, here we utilized a random forest (RF) model with Shapley additive explanations (SHAP) to identify crucial factors influencing aerosol pH and ionic strength, as widely used in previous studies (Li et al., 2024; Zhang et al., 2025). Details of the specifications of RF model can be found in our previous studies (Hong et al., 2022; Fan et al., 2023). Briefly, the RF model was fed by trace gases, meteorological parameters, chemical components and aerosol pH/ionic strength. The inputs were randomly divided into a training set (80%) and a test set (20%). Model performance was evaluated using traversal functions with root-mean-squared error (RMSE), mean-absolute error (MAE), and correlation coefficient ($R^2$) as the evaluation metrics. The contributions of these driving factors were quantified using SHAP values that a larger SHAP value of a feature indicates a higher contribution and the relative importance of various factors can be calculated by the mean of absolute SHAP value. For aerosol pH evaluations, the model input includes T, RH, Weed Speed (WS), boundary layer height (BLH), $PM_{2.5}$, $SO_4^{2-}$, $NO_3^-$, $NH_4^+$, $SO_2$, $O_3$, $NH_3$. It is noted that previous study has point out that RH, as a more general environmental parameter, is more suitable as an input parameter than ALWC for assessing aerosol pH, given the quantitative relationship between pH and ALWC (Zhang et al., 2025). Consequently, ALWC was excluded from the input factors. As shown in **Figure S16a**, the aerosol pH predicted by RF model closely matches our calculation from thermodynamic models, with $R^2$ of 0.96. And RH has the highest contribution (**Figure S16b**) and largest SHAP value (**Figure S16c**) on aerosol pH. Besides, $SO_4^{2-}$ rank as the second most important factors, surpassing the contribution of $NO_3^-$. This result was in consistent with previous work in NCP region (Ding et al., 2019), which declared that $SO_4^{2-}$ had a greater effect than $NO_3^-$ on $PM_{2.5}$ pH as $SO_4^{2-}$ can lead to a higher concentration of $H_{air}^+$ than $NO_3^-$ due to its low volatility and strong dissociation. Furthermore, these outcomes align with correlation analyses, which demonstrate a significant relationship between pH variations and relative humidity (R=-0.48, p<0.05) as well as sulfate (R=-0.37, p<0.05).

[Figure]

**Figure S16**. Performance of the Random Forest (RF) model for predicting aerosol pH (a); the relative importance of each factor for aerosol pH (b); the SHAP (Shapley additive explanations) value of each feature variable on aerosol pH (c). T: Temperature; WS: Weed Speed; BLH: boundary layer height."

**Major Comment #5:**

*General comparisons between HMS (and sulfate, eg page 8) formation to Fairbanks and the urban study in this case are tenuous at best give the vastly different environmental conditions (Fairbanks winter T down to -35C very little sunlight/photochemistry in contrast to this study). These comparisons do not seem valid.*

**Response:** Thanks for the advice. We have removed the comparison with Fairbanks observations and instead concentrated on elucidating the differences in HMS levels between our observations and other regions such as NCP region where cloud/fog formation exerts a more significant influence as detailed in our response to **Major Comment #2.**

**Specific Comments #1:**

*Pg 3 First line, what is meant by "crucial". Add justification (if there is any) for this characterization? Eg, the term "up to" is not specific (ie, it is the max value). What is the mass fraction of HMS in PM2.5 when HMS is 18 g/m3? Compare this to the typical HMS mass fraction in PM2.5? Campbell should not be in the ref list, that is not data from the N. China Plane, check all references here are NCP.*

**Response:** Thanks for the comment. The reason we used "crucial" in the first line is that given the typical ambient levels of other organosulfur compounds, which often fall within the range of ng m$^{-3}$ (Bruggemann et al., 2020), HMS could hold a pivotal role among organosulfur compounds concerning its concentration, especially in NCP region. We acknowledge that in pristine environments (Scheinhardt et al., 2014), ambient HMS levels can be much less significant. Therefore, we have revised the term into "important" in the manuscript and have adjusted our discussion regarding the maximum observed value of 18.5 µg m$^{-3}$. The reference list has also been checked. Please refer to the updated content below.

**Page 3, Line 2**: "In recent decades, hydroxymethanesulfonate (HMS) has emerged as an important organosulfur (OS) species in polluted atmospheres, particularly in northern China (Moch et al., 2018; Song et al., 2019; Ma et al., 2020; Moch et al., 2020; Wei et al., 2020; Liu et al., 2021; Chen et al., 2022; Wang et al., 2024), where peak daily-average HMS concentrations have been measured to reach 18.5 µg m$^{-3}$ (Ma et al., 2020)."

Besides, on the day when the HMS level reached 18.5 µg m$^{-3}$, corresponding to a PM$_{2.5}$ mass concentration of 339 µg m$^{-3}$, the HMS/PM$_{2.5}$ ratio was determined to be 5.6%. This ratio is comparable to findings from studies conducted in NCP region (1−3%) (Wei et al., 2020) and Fairbanks (3−7%) (Campbell et al., 2022), albeit higher than the ratio observed in our investigation (<1%). The lower level of HMS and its reduced fraction within PM$_{2.5}$ in our study may also be attributed to the factors elucidated in our response to **Major Comment #2**.

**Specific Comments #2:**

*Line 10 pg 3, I believe the models of Moch et al are for cloud processing in the formation of HMS. Is this true for all the references here. Please specify, the route, ie clouds/fogs vs aerosol particle water – it matters.*

**Response:** Thanks for the advice. We have removed the comparison with Moch's work and declared the likely minimal contribution of in-cloud/fog HMS formation in urban Nanjing as detailed in our responses to **Major Comment #2**.

**Specific Comments #3:**

*Line 20 pg 3. Regarding meteorological effects. A big influence of temperature is the effect on pH. See Campbell Sci Adv 2024, DOI: 10.1126/sciadv.ado4373,Tao and Murphy ACP 2019 https://doi.org/10.5194/acp-19-9309-2019*

**Response:** Thanks for bring up these papers. We acknowledge that ambient temperature can largely influence the aerosol pH primarily by affecting the partitioning of semi-volatile species (e.g., NH$_4^+$, nitrate and chloride) between gas and aerosol phase (Tao and Murphy, 2019; Campbell et al., 2024) and also by affecting the rate constant of the dissociation reaction of acids to release H$^+$ (Zhang et al., 2025). In contrast to Fairbanks, the temperature in the winter of Nanjing was significantly higher with narrow variations (275±3K). This temperature range was comparable to that during wintertime of Tao and Murphy's observation, which demonstrated that aerosol chemical compositions can also significantly govern the aerosol pH. In this work, we also observed higher pH values under lower temperature condition as shown in the **Figure R1** below. In addition, the random forest model with Shapley additive explanations (SHAP) utilized here as detailed in our reply to **Major Comment #4** also suggests that temperature does affect the aerosol pH, but is less important compared to chemical composition. And this relatively diminished impact of temperature on aerosol pH in this study could be

attributed to its limited variation ranges (275±3K).

[Figure]

**Figure R1**. The relation between aerosol pH and temperature in urban Nanjing.

**Specific Comments #4:**

*Line 36-37 pg 3 that states; In controversy,.... Campbell et al. 2022. I do not agree with this sentence. If you read that paper carefully you will see that pH was estimated since total (gas plus particle) ammonium was not measured. For more discussion on pH in Fairbanks and relation to HMS see Campbell Sci Adv 2024, DOI: 10.1126/sciadv.ado4373*

**Response:** Thanks for the comment. We agree with the reviewer that the pH estimation without NH$_3$ measurements cannot accurately reflect the pH variations in Fairbanks, an environment characterized by extreme cold and localized NH$_3$ emissions. Additionally, we acknowledge that our previous description lacked precision in delineating the complicated relation between aerosol pH and HMS formation in Fairbanks (Campbell et al., 2024). In Fairbanks, initial pH levels can be high, with a relatively high concentration of total ammonium (TA) in comparison to sulfate, especially at extremely low temperatures. This scenario promotes the formation of sulfate and HMS involving SO$_3^{2-}$ reactions. As sulfate and HMS form, the TA/sulfate decreases, leading to a rapid acidification (pH often ranging between -1 and 1), thereby impeding these particulate formation processes. Subsequently, the pH may rise again due to local NH$_{3(g)}$ emissions. Therefore, we have removed the sentence in the Manuscript.

**Specific Comments #5:**

*I also do not understand the line following the one above, line 37-40 page 3. As noted above, why does one assume that the conditions in which the aerosol is being measured represents the conditions under which HMS was formed under. This depends on the estimated lifetime of HMS. This should be considered in the interpretation of these studies and especially the interpretation of the data in this paper.*

**Response:** We agree with the reviewer that HMS detected within aerosols might have originated and transported from prior formation in cloud or fog water, given that the lifetime of HMS could be akin to that of other inorganic aerosols due to its low reactivity that HMS is resistant to oxidation by $H_2O_2$ and $O_3$, and its lifespan against reactions with $\cdot OH$ radicals could exceed two weeks (Lai et al., 2023). Regarding the interpretation of HMS formation in cloud or fog water and its implications on our observations, we kindly direct you to our response provided in **Major Comment #2** for a detailed discussion on this matter.

**Specific Comments #6:**

*Line 13 page 4. To have ALWC of 100's of ug/m3 in polluted urban areas it must correspond to very poor air quality only found in some countries – ie this is not typical. Be specific what urban areas would have this level of ALWC, ie, in cities in the NCP?*

**Response:** Thanks for the comment. Except for NCP region, other countries in Asia such as Indian and Nepal also experiences severe air pollution events during the winter season, with maximum $PM_{2.5}$ values exceeding 300 $\mu g \ m^{-3}$ (Islam et al., 2020; Nair et al., 2020). And the ALWC has been reported to be hundreds of $\mu g \ m^{-3}$ across different cities in Indian (Gopinath et al., 2022). Here we also changed the sentence in the Manuscript.

**Page 4, Line 12:** "For instance, the aerosol LWC can exhibit a wide range, from tens of $\mu g \ m^{-3}$ in the marine atmosphere to hundreds of $\mu g \ m^{-3}$ under severe Asia haze (Gopinath et al., 2022)."

**Specific Comments #7:**

*First lines of page 6. It makes no sense using the epsilon of 0.5 for NH3/NH4+ partitioning in the pH analysis based on Fairbanks Alaska data which is an extremely different environment than that of this study. One might consider an iterative approach, see doi:10.2533/chimia.2024.1*

**Response:** Thanks for bringing this to our attention. It appears there was an inadvertent citation error in our previous description. The $\varepsilon (=NH_4^+/(NH_3+NH_4^+))$ value referenced in our work was derived from observations conducted in the Bohai Sea in May 2021 (Wang et al., 2022a), where the $NH_3$ concertation averaged at 3.4±1.6 $\mu g \ m^{-3}$ and the $NH_4^+$ level averaged at 4.0±3.7 $\mu g \ m^{-3}$. Despite observing lower $NH_4^+$ levels during our Bohai Sea and Huanghai Sea cruise, we opted to utilize this $\varepsilon$ value due to the resemblance of our observational environment in April 2023 to that of the aforementioned study. We have corrected this citation in the Manuscript.

**Page 6, Line 2:** "For marine aerosols, we adopted a particle to particle + gas partitioning fraction of $NH_4^+$ ($\varepsilon$) of 0.5, drawing from prior observations conducted in

the Bohai Sea in May (Wang et al., 2022) in light of the absence of gaseous data."

**Specific Comments #8:**

*R1 and R2 are shown as equilibrium reactions, but then the reaction rate constants are discussed. This is confusing.*

**Response:** Thanks for the comment. These reactions (**R1**, **R2**) are reversible, and the first-order rate constant of HMS decomposition ($k_d$) have been measured in several laboratory experiments showing the decomposition of HMS was so slow under acidic condition that the chemical equilibrium is difficult to achieve (Boyce and Hoffmann, 1984; Deister et al., 1986; Kok et al., 1986). For instance, Kok et al. (1986) measured the $k_d$ of $3.5 \times 10^{-6}$ s$^{-1}$ at pH of 5. Assuming a gas-phase $SO_2$ level of 1 ppb and HCHO of 1 ppb, the general formation rate of HMS ($R_f$, M s$^{-1}$) in dilute solution is calculated to be $2.5 \times 10^{-4}$ M s$^{-1}$ at temperature of 273 K. Considering that the HMS concentration in cloud/aerosol water were far below 1 M, the decomposition rate of HMS could be at least two orders of magnitude lower than its formation at pH of 5. Although both HMS formation rate and $k_d$ both increase with higher pH values, model study suggested that the decomposition of HMS was insignificant and HMS levels are predominantly controlled by formation kinetics in the typical pH range of cloud/fog and aerosol water (pH < 6) based extrapolations of $k_d$ measurements (Song et al., 2021). We have added above information into the manuscript.

**Page 6, Line 15:** "These reactions are reversible, and the first-order rate constant of HMS decomposition have been measured in previous studies showing the decomposition of HMS was so slow that HMS levels are predominantly controlled by formation kinetics in the typical pH range of cloud/fog and aerosol water (pH < 6) (Boyce et al., 1984; Deister et al., 1986; Kok et al., 1986; Song et al., 2021)."

**Specific Comments #9:**

*Fig 1 caption, typo change penal to panel. T in (a) is in C, t in (e) is in Kelvin, why? Why is HMS not shown in Fig 1d?*

**Response:** Thanks for pointing it out. We have corrected the typo and change the T in (a) to the unit of Kelvin. And HMS have also been included in **Figure 1d**. Please checked our updated **Figure 1** in the Manuscript.

[Figure]

**Figure 1.** Left panel (Urban Observations): Time series of atmospheric measurements in urban Nanjing, including temperature and relative humidity (a); $SO_2$ and HCHO levels (b); HMS and sulfate concentration (c); $PM_{2.5}$ composition (d). The gray shade area highlighted a 7-days' haze episode (December 27[th], 2023 to January 2[nd], 2024) with $PM_{2.5}$ mass concentration exceeding 75 μg m[−3]. Other ancillary measurements in urban atmosphere including wind speed and wind direction, $PM_{2.5}$ mass concentration, $O_3$, $NH_3$ level, fog occurrence and cloud water content can be found in **Figure S2**. Right panel (Marine Observations): Atmospheric measurements during marine cruise, including temperature and relative humidity (e); $SO_2$ (f), HMS and sulfate level where the biogenic nss-sulfate (bio-nss-sulfate) was calculated based on previously measured bio-nss-sulfate/nss-sulfate ratio of 0.14 (Yang et al., 2015) (g); $PM_{2.5}$ composition (h).

**Specific Comments #10:**

*Last line page 7, referring to similar results from Moch et al. But wasn't Moch's results based on HMS formation in clouds, which is not what is assumed here? Also, next line on top of page 8, comparing HMS concentrations to Fairbanks makes little sense given how different the situation is.*

**Response:** Thanks for the comment. We agree with the reviewer that comparisons between our study and Moch's work may not be appropriate, as their study focused on cloud HMS formation. Likewise, comparing HMS observations in Fairbanks poses challenges due to the their extremely low temperatures and its consequential effects on pH levels and heterogeneous chemistry. Thus, we have removed the comparison and instead concentrated on elucidating the differences in HMS levels between our observations and other regions where cloud/fog formation exerts a more significant influence as detailed in our response to **Major Comment #2.**

**Specific Comments #11:**

*Line 40 page 8, too strong a statement, correlation is not causation. This applies throughout the manuscript.*

**Response:** We agree with the reviewer. And the statement has been changed in the Manuscript. And as for the driving factors for aerosol pH variations, we have replaced the supports from correlation analysis with a random forest (RF) model combined Shapley Additive Explanations with details given in our reply to **Major Comment #4**.

**Page 9, Line 29:** "Organic components in urban aerosols, particularly WSOC, also exhibited a strong correlation with sulfate and HMS levels (**Figure S4**) during our observations, probably indicating their likely involvement in aqueous sulfur chemistry processes."

**Specific Comments #12:**

*Fig 2 b (and in other places) specify if ratio of HMS/Sulfate is mass or molar.*

**Response:** Thanks for the hint. The ratio is molar ratio. We also specify it in other places.

**Specific Comments #13:**

*Page 10 top of page, what is the lifetime of HMS, this would support the idea that it is formed locally. Low wind speed does support the idea of local formation but does not prove the conditions measured were those under which HMS was formed.*

**Response:** Thanks for the comment. The lifetime of HMS could be akin to that of other inorganic aerosols. This is supported by the fact that HMS exhibits low reactivity, rendering it resistant to oxidation by $H_2O_2$ and $O_3$, and its lifespan against reactions with $\cdot OH$ radicals could exceed two weeks (Lai et al., 2021). We agree with the reviewer that HMS detected within aerosols might have originated or transported from prior formation in cloud or fog water. Discussion on this matter and our edition in the Manuscript, please refer to our response to **Major Comment #2**.

**Specific Comments #14:**

*Fig 3a. I assume this graph is predicted IS vs measured RH; this should be clarified. IS depends on ALWC and ALWC depends on RH and other factors (particle composition). Why not focus on ALWC and IS, that is what matters and what is more generalizable to other locations vs RH.*

**Response:** Thanks for the question. We agree with the reviewer that ALWC depends on RH and other factors such as particle composition, with RH being the most influential factor on the variation of ALWC (Gopinath et al., 2022). Higher RH levels

enable increased water absorption, subsequently elevating ALWC and diluting the aerosol solution. In the meantime, heightened ALWC can in turn facilitate the formation of inorganic ions through gas-particle conversion and the partitioning of gas pollutants. This process can potentially lead to an accumulation of aqueous-phase ionic components, complexing the relationship between ALWC and ionic strength. Here we also utilized a random forest (RF) model with Shapley Additive Explanations to identify crucial factors influencing aerosol ionic strength in real atmosphere with details given in our reply to **Major Comment #4**. To determine the key drivers for ionic strength variations, the model input includes T, RH, Weed Speed (WS), boundary layer height (BLH), $PM_{2.5}$, $SO_4^{2-}$, $NO_3^-$, $NH_4^+$, $SO_2$, $O_3$, ionic strength and ALWC. As shown in **Figure S16d**, the aerosol ionic strength predicted by RF model closely matches the outputs of thermodynamic models, with $R^2$ of 0.95. And RH has the dominant contribution of 76% and largest SHAP value on aerosol ionic strength, following by the ALWC (**Figure S16e-f**). These findings align with correlation analyses that demonstrated a stronger correlation between RH and IS (R=-0.89) compared to that between ALWC and IS (R=-0.62), underscoring the predominant influence of RH in determining ALWC and IS in our study area.

Moreover, while uncertainties may persist in estimating ALWC using diverse methods and assumptions (e.g., with or without considering organics), RH serves as a readily accessible and accurate parameter. Hence, we would like to maintain the comparison between RH and predicted IS. Additionally, a modeling study has also reported a linear IS response to RH within the typical humidity range of haze events (60-90%), although the specific regression was not provided (Song et al., 2018). Notably, despite that NCP region exhibiting elevated pollution levels ($PM_{2.5} > 400$ μg $m^{-3}$) and ALWC reaching levels in mg $m^{-3}$ range during humid hazy days (RH > 80%), the estimated IS range (5.6±2.9 mol/kg) aligns with our findings as depicted in **Figure R2** (Wang et al., 2024). However, IS predictions under lower humidity were higher compared to our study. This might due to two factors: 1) the neglecting of organic species in their thermodynamic models leading to underestimation in ALWC and 2) the more persistent accumulation of ionic species under lower humidity level (RH<60%) as indicated by the much higher $PM_{2.5}$ concentrations (i.e., >75 μg $m^{-3}$) compared to our observations. Take together, despite that the regression between RH and IS may not be universally applicable to other region, **Figure 3a** effectively illustrates the correlation between RH and predicted ionic strength within our study. Given that haze events are commonly linked to elevated RH levels, our current emphasis is on underscoring the reduction in IS levels under increased humidity, rather than concentrating on the specific regression between RH and IS. Following description regarding the factors determining the variations aerosol ionic strength observed in winter of Nanjing has been included in the Manuscript and Supporting Information.

**Revision in Manuscript:**

**Page 11, Line 37:** "While acknowledging that the our regression between RH and IS may not be universally generalizable to other regions, our findings are consistent with earlier field and model research indicating a decline in aqueous ionic concentrations on

humid haze days due to the rapid rise in ALWC (Song et al., 2018; Shen et al., 2024a; Wang et al., 2024)."

**Revision in Supporting Information:**

**Page 18, Line 12:** "To determine the key drivers for ionic strength variations, the model inputs include T, RH, WS, BLH, PM$_{2.5}$, SO$_4^{2-}$, NO$_3^-$, NH$_4^+$, SO$_2$, O$_3$ and ALWC. As shown in **Figure S16d**, the aerosol ionic strength predicted by RF model closely matches the outputs of thermodynamic models, with R$^2$ of 0.95. And RH has the dominant contribution of 76% and largest SHAP value on aerosol ionic strength, following by the ALWC (**Figure S16e-f**). This could be explained by that higher RH levels enable increased water absorption, subsequently elevating ALWC. In the meantime, heightened ALWC can facilitate the formation of inorganic ions through gas-particle conversion and the partitioning of gas pollutants, potentially leading to an accumulation in the aqueous-phase ionic components. These findings align with correlation analyses that demonstrated a stronger correlation between RH and Ionic Strength (IS) (R=-0.89) compared to that between ALWC and IS (R=-0.62), underscoring the predominant influence of RH in determining ALWC and IS in our study area.

[Figure]

**Figure S16**. Performance of the Random Forest (RF) model for predicting aerosol pH (a) ionic strength (d); the relative importance of each factor for aerosol pH (b) and ionic strength (e); the SHAP (Shapley additive explanations) value of each feature variable on aerosol pH (c) ionic strength (f). T: Temperature; WS: Weed Speed; BLH: boundary layer height."

[Figure]

**Figure R2.** The relation between aerosol ionic strength in this work (including both marine and urban data) and from NCP region (data adopted from Wang et al., 2024) where the ionic strength values were estimated without the consideration of organics.

**Specific Comments #15:**

*Fig 4b, the colors for data, urban clean vs polluted are too close to being the same. Typo in caption (radium?). From this plot, does the data fit with the model? Looking at the urban polluted vs clean, it does not seem to fit that well, there are many blue markers on similar isopleths as urban clean and urban polluted. The graph does not actually show data on measured HMS formation rates, or maybe all one can do is show HMS concentration. So how does this plot assess the comparison between these data from this study and the predictions. At the very least state associated HMS concentrations for the groups of data shown on the plot. The data on HMS concentrations and production rates of HMS, eg Fig 2, should be added. Furthermore, the plot is for ALWC = 100 ug/m3, but the data seem to cover a wide range of ALWC. How is that reasonable?*

**Response:** Thanks for the comment. The original **Figure 4b** consisted of a $P_{HMS}$ isopleths figure under different pH and IS level, accompanied by data points representing pH and IS values from different environmental conditions (i.e., urban haze, urban clean and marine atmospheres). The $P_{HMS}$ isopleths figure was intended to specifically compare the role of pH and IS in affecting the HMS formation rates, thus we used constant value for other parameters such as precursor concentration, temperature and ALWC. However, we acknowledge that while these dots represent the pH and IS levels for clean, hazy, and marine aerosols, their corresponding formation rates in the isopleths may not precisely mirror the actual $P_{HMS}$ due to disparities in other variables. To avoid confusion, we have separated original **Figure 4b** into three subfigures (**Figure 4b-d**) to specifically represent the interplay between aerosol acidity and ionic strength under different environmental setting (i.e., urban haze, urban clean and marine atmospheres), by using their characteristic conditions as summarized in **Table S1**. The comparison between revised **Figure 4b** and **Figure 4d** clearly shows that the $P_{HMS}$ values during clean period in urban Nanjing, characterized by elevated pH and IS values, were generally 1 to 2 orders of magnitude lower than those observed

during polluted days, underscoring the role of reduced ionic strength level under higher humidity in promoting HMS formation. And lower $P_{HMS}$ ($10^{-5}$–$10^{-4}$ µg m$^{-3}$ h$^{-1}$) were observed for marine aerosols (**Figure 4c**). The results were consistent with our $P_{HMS}$ calculations.

Besides, it was observed that the variations in other factors such as precursors' concentration and ALWC have no impact on the location of ridgelines (**Figure 4b-d**), which represents the IS level associated with the highest $P_{HMS}$ at a given pH. Above the ridgeline, a pH-limited regime can be identified, where HMS production rate escalated with increasing pH but diminished with rising IS levels. Conversely, below the ridgeline, a probable co-limited regime emerged, where $P_{HMS}$ rose with both IS and pH levels. This result highlights the more pronounced impact of aerosol pH and ionic strength on regulating HMS formation compared to other parameters. We have also made revision in the Manuscript and **Figure 4**.

**Revision in Manuscript:**

[revised manuscript text omitted]

Regarding the comparison between predicted $P_{HMS}$ and ambient HMS level, a previous study attempted to correlate $P_{HMS}$ ($\mu g\ m^{-3}\ h^{-1}$) and daily averaged HMS levels

by multiplying 24 hours with $P_{HMS}$ as a rough approximation (Shen et al., 2024a) given the low reactivity of HMS after its formation although other physical process such as diffusion and deposition can also impact its ambient concentration (Liu et al., 2024). As shown in **Figure R3**, our comparison between predicted HMS and measured HMS levels exhibited a consistent pattern, with higher concentrations during haze events. The averaged HMS level predictions during hazy days were in line with ambient observations, whereas the estimated HMS level was nearly an order of magnitude lower than the measurements on clean days. This discrepancy can be largely attributed to the fact that ambient HMS levels are influenced by various key atmospheric processes, including physical transport (advection and turbulent diffusion), chemical reactions, and deposition losses, whereas $P_{HMS}$ solely represents the averaged rate for HMS formation process. While we acknowledge that chemical transport models offer a more comprehensive representation of these atmospheric processes and can provide more precise estimations of HMS levels, the primary focus of our $P_{HMS}$ calculations was to emphasize the enhanced HMS formation during hazy days, as evidenced by the higher HMS/CO ratio (as shown in **Figure 2f**, attached below), and the significance of aerosol ionic strength in this context. Despite potential uncertainties may persist in $P_{HMS}$ calculations reported here due to the lack of laboratory-based kinetics under higher ionic strength level (i.e., IS > 11 mol kg$^{-1}$) and potential enhancement of HCHO solubility within aerosol water as detailed in the Manuscript (Page 14, Line 35), these $P_{HMS}$ values were intended as an initial approximation to depict the potential fluctuations of HMS formation rates under different pollution conditions, rather than for direct comparison with ambient HMS levels.

[Figure]

**Figure R3.** Comparisons between observed HMS level and predictions using HMS formation rates.

[Figure]

**Figure 2f.** The comparison between HMS/CO ratio and HMS formation rates ($P_{HMS}$). There the HMS/CO ratio was used to better represent the secondary formation of ambient HMS as CO was usually considered to be an inert chemical species during rapid haze formation, with its variation often interpreted as indicative of the accumulation of primary pollutants in the shallower boundary layer (Williams et al., 2016).

**Specific Comments #16:**

*I do not understand the first part of the last paragraph, how does cycling of HMS through formation and loss in the liquid drop make sulfate? Explain more.*

**Response:** Thanks for the question. Here we prepared a schematic diagram to help represent this process (**Scheme 1**). This cycling was initiated by oxidation of HMS by $^\bullet$OH, releasing the aqueous-phase HCHO ($HCHO_{(aq)}$) and sulfite radical ($SO_3^{\bullet-}$), which was quickly oxidized into peroxysulfate radical ($SO_5^{\bullet-}$). These $SO_5^{\bullet-}$ radicals can engage with further reactions, ultimately resulting in sulfate formation (Seinfeld and Pandis, 2016; Lai et al., 2023). Model simulations indicate that this reaction can lead to a reduction of atmospheric concentrations of HMS by 10-20% in winter and 40-60% in summer on a global scale (Song et al., 2021). Due to the lower atmospheric abundance of HMS compared to $SO_4^{2-}$, the contribution of above reaction to $SO_4^{2-}$ is usually ignored in global/regional chemical transport models. However, the $HCHO_{(aq)}$, formed alongside the formation of $SO_5^{\bullet-}$, can stay in aqueous phase due to its high solubility and subsequently react with $SO_{2(aq)}$ to reform HMS and then be released back into water solution during the oxidation of HMS by $^\bullet$OH. Although dissolved $SO_{2(aq)}$ and $^\bullet$OH compete the $HCHO_{(aq)}$ formed from HMS oxidation, reaction between $HCHO_{(aq)}$ and $SO_{2(aq)}$ may be an important reaction pathways for $HCHO_{(aq)}$ in ambient aerosols, given the much lower $^\bullet OH_{(aq)}$ concentration in cloud/aerosol water ($[^\bullet OH_{(aq)}]= 10^{-15}$-$10^{-16}$ M) (Herrmann et al., 2015) and the largely enhanced reaction rate between $HCHO_{(aq)}$ and $SO_{2(aq)}$ due to the high ionic strength in aerosol water (Zhang et al., 2023). Under such circumstance, HMS can be considered as atmospheric sulfur reservoir with $HCHO_{(aq)}$ acting as the catalyst that facilitates the continuous conversion from gas-phase $SO_2$ to $SO_4^{2-}$. We have added more explanation in the Manuscript.

**Page 19, Line 5:** "In addition, laboratory studies have reported that although HMS is resistant to oxidation by $H_2O_2$ or $O_3$, it can be oxidized by $^\bullet OH$ through aqueous or heterogenous reactions, eventually leading to the formation of $SO_4^{2-}$ after a series of radical chain reactions (Seinfeld et al., 2016; Lai et al., 2023). Model simulations also indicate that this reaction can lead to a reduction of atmospheric concentrations of HMS by 10-20% in winter and 40-60% in summer on a global scale (Song et al., 2021). More importantly, model studies underscore the potentially critical role of HMS as an atmospheric sulfur reservoir in sulfate formation (Dovrou et al., 2022; Zhao et al., 2024). This significance arises from the fact that the $HCHO_{(aq)}$ can be released during the oxidation of HMS, which can react with $SO_2$ to regenerate HMS and be subsequently released back into the aqueous phase during further HMS oxidation by $^\bullet OH$. Under such circumstance, HMS can be considered as atmospheric sulfur reservoir with $HCHO_{(aq)}$ acting as the catalyst that facilitates the continuous conversion from gas-phase $SO_2$ to $SO_4^{2-}$. A recent modeling work indicated that this $HCHO_{(aq)}$ catalysis process could contribute up to 20-30% of particulate sulfur formation under coastal and marine conditions (Zhao et al., 2024), despite that uncertainties may persist due to the lack of laboratory-based kinetic parameters for HCHO-related pathways and the oversight of IS effects in HMS and sulfate formation."

[Figure]

**Scheme 1**. Schematic representation of the HCHO catalytic processes on $SO_4^{2-}$ formation during heterogenous $^\bullet OH$ oxidation of HMS.

**Specific Comments #17:**

*Page 17, what is Enteromorpha?*

**Response:** Enteromorpha is a kind of marine algae, which can release formaldehyde during photodecomposition process. We attached a picture of Enteromorpha for the reviewer's reference.

[Figure]

**Figure R4**. The collected Enteromorpha sample (Shen et al., 2024b).


**Anonymous Referee #2**

*The manuscript presents a detailed study of HMS formation in urban and marine environments, examining the occurrence of a haze event in urban Nanjing. An ion chromatography method is used to analyze collected samples for HMS quantification and ISORROPIA II and AIOMFAC were used to estimate ALWQ, pH and ionic strength (IS). An interesting effect of the ionic strength in HMS formation is discussed, providing evidence of HMS enhancement at low ionic strength during high humidity and pollution days. Although the study is well structured it lacks support of some key results, raising questions regarding the pH estimation, and model results vs measurements. The results are interesting, and overall, the work is suitable for publication in "Atmospheric Chemistry and Physics (ACP)" provided that the comments outlined below are adequately addressed.*

**We thank the reviewer for his/her thoughtful comments. The referee's comments are below in italics followed by our responses in normal font.**

**Comment #1:**

*The authors use hydrogen peroxide to react the available S(IV) species, since the ion chromatography signal presents HMS as sulfite. Although hydrogen peroxide is not expected to decompose HMS at pH<6, have the authors tested this reaction for the examined conditions? Since filter aliquots are used the pH needs to monitored and test samples need to be examined for potential HMS decomposition (even from formed radicals) in the analyzed sample.*

**Response:** Thanks for the advice. We have tested the pH of filter aliquots of two hazy days (December 28$^{th}$ and 31$^{st}$, 2023) and one clean day (December 24$^{th}$, 2024) using a pH-meter (Sartorius, PB-10). These aliquots were extracted following the same procedure as described in the **Section S1** (Supporting Information) that a 16 mm punch of each filter was cut and extracted using 10 mL of Millipore water (18.2 MΩ). The selection of these specific days was predicated on their comparatively elevated aerosol pH levels. The determined pH values were 5.68, 5.71, and 5.79, respectively, all falling below the threshold of 6. Besides, half of these aliquots were immediately injected to IC and the other half was injected after 3 hours. The results shown there was less than 5% of difference in HMS level between these IC analyses. Therefore, here we believe the potential decomposition of HMS could be negligible. Following information was added into the **Section S1** in Supporting Information.

Page 10, Line 35: "Considering that HMS is quite stable under acidic conditions (pH<6) but become unstable under alkaline environments and dissociates rapidly into $SO_3^{2-}$

and HCHO (Seinfeld and Pandis, 2016), here we also tested the pH of filter aliquots of two hazy days (December 28[th] and 31[st], 2023) and one clean day (December 24[th], 2024) using a pH-meter (Sartorius, PB-10). The determined pH values were 5.68, 5.71, and 5.79, respectively, all falling below the threshold of 6. And half of these aliquots were immediately injected to IC and the other half was injected after 3 hours. The results shown there was less than 5% of difference in HMS level between these IC analyses."

**Comment #2:**

*The study focuses mainly on aerosol HMS presence, without providing evidence that the HMS formation occurred in the aerosol vs the cloud/fog water phase. Since 66% of HMS is formed in cloud and fog water, an analysis and/or discussion on the fate and formation of HMS in cloud and fog water is required and a comparison with this studies finding in the aerosol phase is necessary.*

**Response:** Thanks for raising this critical point. We agree with the reviewer that cloud could serve as a significant medium for HMS formation due to the large amount water content (CWC). It is noted that 66% of in-cloud HMS formation was a general value during their observation period from November 26[th] to December 16[th], 2015, Beijing (Wang et al., 2024). However, near-surface aerosol HMS formation (HMSp) was found to be comparable to those formed in cloud/fog water (HMSc) during their Episode 1. Moreover, under certain days (e.g., November 28[th] to 29[th]), HMSp levels could exceed HMSc level, which was attributed to that the enhancing effect of IS can counterbalance the constraints imposed by lower aerosol water content and pH values compared to cloud water. However, the exact extent and scope of this compensatory mechanisms remain ambiguous. In addition, another work has reported the detection of HMS in aerosols regardless of the presence of cloud/fog in the winter of Beijing, and HMS concentrations were found to show a good correlation (R= 0.92, P< 0.01) with aerosol liquid water content (ALWC) (Ma et al., 2020). Therefore, they proposed that aerosol water can serve as a medium for HMS formation despite the fact that HMS levels were significantly lower ($< 1$ $\mu g$ $m^{-3}$) in the absence of cloud/fog and peak HMS level of 18.5 $\mu g$ $m^{-3}$ was observed during fog processes. Furthermore, previous study has emphasized that although in-cloud formation processes can be much faster compared to aerosol water given its large water content, their contribution to the near-surface aerosol composition can be negligible during winter haze in Northern China due to the accompanied temperature inversions and high atmospheric stability so that the weak vertical exchange impeded the transportation of gas precursors emitted near surface to high altitudes and the chemicals produced in the high-altitude cloud could not be easily transported to the ground (Wang et al., 2022).

In our current work, no prolonged fog events (RH > 90% and Visibility < 1 km) lasting over two hours were observed as shown in **Figure S2e**, which was attached below for the reviewer's reference. Here, we also utilized the vertical temperature profile obtained from http://weather.uwyo.edu/upperair/sounding.html to identify temperature inversion in the atmospheric boundary layer in the winter of Nanjing. As

shown in **Figure S3**, a predominant occurrence of temperature inversions was noted across our observation days, characterized by inversion layers extending from the surface to altitudes of 2000 m, particularly evident on hazy days. Besides, days without temperature inversions (i.e., from December 19[th] to December 23[th], 2023) were associated with negligible cloud water content (**Figure S2f**), which was obtained from MERRA-2 (Modern-Era Retrospective analysis for Research and Applications, Version 2) (Gelaro et al., 2017). In line with Ma et al.'s work (2020), we also detected HMS within $PM_{2.5}$ samples during 7 of non-cloud days. Besides, our data indicated no discernible correlation between CWC and HMS levels whereas HMS levels displayed a notably positive correlation with ALWC (R=0.67, P<0.01). While we recognize that the HMS formation in cloud layers characterized by persistent dense clouds and extended durations may carry great contributions, **in the context of this study, the presence of temperature inversions, low wind speeds, and diminished cloud water content during the winter season in Nanjing suggests that the impact of in-cloud HMS formation on the observed levels of particulate HMS may be insignificant**. **Section 3.2** has been extensively revised to provide a more detailed explanation of the role of in-cloud formation in observed particulate HMS levels during our winter observations in Nanjing. We have also made edits in the **Abstract**.

**Revision in Abstract:**

**Page 2, Line 1:** "Hydroxymethanesulfonate (HMS) has emerged as a critical organosulfur species in ambient aerosols, yet the impact of aerosol properties, particularly ionic strength (IS), on the formation of HMS remains uncertain. Here, HMS levels in wintertime of urban Nanjing, China were quantified at 0.30±0.10 μg m$^{-3}$, where the contribution of in-cloud formation likely carry minor significance due to the barrier resulted from stable stratification."

**Revision in Manuscript:**

**Page 8, Line 1:** "

[revised manuscript text omitted]

**Revision in Supporting Information: "**

[Figure]

**Figure S2.** Ancillary atmospheric measurements in urban Nanjing including fog event criteria based on hourly relative humidity (RH>90%) and Visibility (<1 km) data (e); Time series of the average cloud water content below the planetary boundary layer height over our observation sites (f).

[Figure]

**Figure S3**. Observed vertical temperature profile at 00 UTC during our observation in Nanjing. The dates of hazy days were marked by red.

As for marine environment, we attempt to determine the HMS formation rate in cloud water ($P_{HMS, c}$) and aerosol water ($P_{HMS,a}$) considering the minimal impact of temperature inversion during our observation in April. In the manuscript, we have compared the $P_{HMS, a}$ ($2.06 \times 10^{-5}$ µg m$^{-3}$ h$^{-1}$) in our study with $P_{HMS, c}$ ($1.2 \times 10^{-5}$ µg m$^{-3}$ h$^{-1}$) estimated for observations near Beaufort Sea with a CWC of 4.8 mg m$^{-3}$ and a pH of 5 (Liu et al., 2021). Using these same CWC and pH, here we calculated the averaged $P_{HMS,c}$ of $4.95 \times 10^{-5}$ µg m$^{-3}$ h$^{-1}$ in our marine environment, which was comparable to the $P_{HMS, a}$. The calculation results and input parameters were summarized in **Table R1**. This comparable $P_{HMS,a}$ and $P_{HMS,c}$ can be explained by that despite that the CWC being ~ 300 times higher than ALWC and aerosol pH being lower, the higher ionic strength level in marine aerosol water can significantly promote HMS formation process, rendering the HMS formation within aerosol water a process comparable to that observed in cloud and fog environments. Altogether, although these formation rates serve as initial approximations, **these results may indicate the significant contribution of both aerosol and cloud water in the formation of HMS in marine atmosphere**. Following discussion has been included in **Section 3.4**.

**Page 17, Line 10:** "Using the averaged [SO$_{2(g)}$]=0.82 ppb and [HCHO$_{(g)}$]=0.5 ppb (Ervens et al., 2003; Zhao et al., 2024), the potential HMS formation rates in these marine aerosols were calculated to be $2.06 \times 10^{-5}$ µg m$^{-3}$ h$^{-1}$, a value comparable to reported cloud/fog-based HMS production rate ($1.2 \times 10^{-5}$ µg m$^{-3}$ h$^{-1}$) near Beaufort Sea (Liu et al., 2021) with a liquid water content (LWC) of 4.8 mg m$^{-3}$ and a pH of 5. Considering the minimal impact of temperature inversion during our observation in April, here we also determined the in-cloud HMS formation rate to be $4.95 \times 10^{-5}$ µg m$^{-3}$ h$^{-1}$ in our marine environment using a LWC of 4.8 mg m$^{-3}$ and a pH of 5. These results further highlighted the significance of IS-dependent enhancement in HMS formation, which can render the HMS formation within aerosol water a process comparable to that observed in cloud and fog environments, particularly in humid and pristine conditions."

Besides, for the reviewer's reference, here we also attempt to compared the HMS formation rate in cloud water ($P_{HMS, c}$) and aerosol water ($P_{HMS, a}$) in urban Nanjing although the transportation of in-cloud HMS formation can be largely impeded. Global model (Song et al., 2019) has estimated the cloud pH in the winter of Nanjing aera to be around 4-5, consistent with winter observations in China (4.1±0.6) (Shah et al., 2020). In our analysis, we utilized the upper value of cloud pH of 5 to estimate $P_{HMS, c}$ under pre-haze and haze conditions, comparing it with $P_{HMS, a}$. Despite the tendency for cloud pH to decrease during haze pollution (Li et al., 2017), it was observed that $P_{HMS, c}$ was five times lower than $P_{HMS, a}$ on hazy days (**Table R1**). Nevertheless, it is worth noting that the $P_{HMS, a}$ value of $1.06\pm0.64 \times 10^{-2}$ µg m$^{-3}$ h$^{-1}$ can effectively reflect the HMS levels observed in ambient aerosol, furthering indicate the predominate role of aerosol water for HMS formation during temperature inversion. Besides, the $P_{HMS, c}$ were found to be comparable to $P_{HMS, a}$ on clean days. Considering possible overestimations in $P_{HMS, c}$ due to reduced precursor concentrations and hindered transport to ground level resulting from weak vertical exchange, these results suggest

the limited impact of in-cloud HMS formation on detected HMS levels. We acknowledge that uncertainties may exist in the calculation of HMS formation rate due to potential variability associated with cloud pH, thus above information was not included in the Manuscript.

**Table R1**. Summary of $P_{HMS, a}$ and $P_{HMS, a}$ as well as input parameters.

| | Marine | | Urban | | | |
|---|---|---|---|---|---|---|
| | **Cloud** | **Aerosol** | **Cloud** | | **Aerosol** | |
| | | | Pre-haze | Haze | Pre-haze | Haze |
| T (K) | 283±1 | 283±1 | 272±2 | 277±4 | 272±2 | 277±4 |
| SO$_2$ (ppb) | 0.82±0.42[a] | 0.82±0.42 | 1.99±0.45 | 2.21±0.69 | 1.99±0.45 | 2.21±0.69 |
| HCHO (ppb) | 0.5[a] | 0.5 | 1.91±1.05 | 5.38±1.08 | 1.91±1.05 | 5.38±1.08 |
| LWC ($\mu$g m$^{-3}$) | 4.8[b] mg m$^{-3}$ | 14.72 | 1.5±2.2 mg m$^{-3}$ | 5.12±5.45 mg m$^{-3}$ | 22.69±20.65 | 80.45±38.40 |
| pH | 4.8[b] | 4.24±0.15 | 5[d] | 5 | 5.25±0.42 | 4.76±0.46 |
| Ionic strength (mol kg$^{-1}$) | 10$^{-4}$[c] | 4.30±1.24 | 10$^{-4}$ | 10$^{-4}$ | 12.32±3.19 | 8.85±1.30 |
| P$_{HMS}$ ($\times$ 10$^{-4}$ $\mu$g m$^{-3}$ h$^{-1}$) | 0.51 | 0.21 | 2.56±4.68 | 20.4±23.3 | 6.40±6.96 | 106±64.6 |

[a] The levels of SO$_2$ and HCHO over marine environments were adopted from previous studies (Ervens et al., 2003; Zhao et al., 2024).
[b] The LWC and pH for cloud/fog droplets came from previous coastal study near Beaufort Sea (Liu et al., 2021).
[c] This typical ionic strength value of cloud droplet was given by Herrmann et al. (2015).
[d] This cloud pH was the upper value of global estimation on Nanjing aera (Song et al., 2019).

**Comment #3:**

*The estimation of pH is achieved without ammonia measurements. How accurate is this estimation? More information is needed.*

**Response:** Thanks for the comment. To clarify, we monitored the levels of gas-phase ammonia during our observations in Nanjing using a Monitor for AeRosols and GAses (MARGA; Metrohm Ltd., Switzerland) to estimate urban aerosol pH as detailed in **Section 2.1** (Page 5, Line 15). The daily averaged NH$_3$ levels were added into Supporting Information as **Figure S2c** (attached below). For marine aerosol, we utilized the $\varepsilon$(=NH$_4^+$/(NH$_3$+NH$_4^+$)) value from previous study which conducted their observations in the Bohai Sea in May 2021 (Wang et al., 2022), where the NH$_3$ concertation averaged at 3.4±1.6 and the NH$_4^+$ level averaged at 4.0±3.7 $\mu$g m$^{-3}$. Considering the resemblance of our observational environment in April 2023 in Bohai Sea and Huanghai Sea to that of the referenced study, we used a $\varepsilon$ value of 0.5 to determine the marine aerosol pH. We would like to declared that there was an inadvertent citation error in our previous reference and we have corrected this in the Manuscript.

**Page 6, Line 3:** "For marine aerosols, we adopted a particle to particle + gas partitioning fraction of $NH_4^+$ (ε) of 0.5, drawing from prior observations conducted in the Bohai Sea in May (Wang et al., 2022) in light of the absence of gaseous data."

[Figure]

**Figure S2.** Ancillary atmospheric measurements in urban Nanjing including gas-phase $NH_3$ and $HNO_3$ level (c).

**Comment #4:**

*The model results are interesting, providing new insights on HMS fate. However, with the exception of correlations, how do these results compare with the measurements? What are the most sensitive parameters of the model and what is the deviation between measurements and model results. How does this affect the conclusions?*

**Response:** Thanks for the comment. As shown in **Figure 2** (attached below), this work estimated larger HMS formation rates ($P_{HMS}$) during haze events compared to clean days (**Figure 2f**), showcasing a consistent trend with ambient HMS level measurements (**Figure 2c**). In addition, considering other atmospheric process such as physical transport could also impact ambient HMS level, we has utilized the HMS to carbon dioxide (CO) ratio to better represent the secondary formation of ambient HMS as CO was usually considered to be an inert chemical species during rapid haze formation, with its variation often interpreted as indicative of the accumulation of primary pollutants in the shallower boundary layer (Williams et al., 2016). Therefore, the larger HMS/CO ratio observed during haze events further evidenced the enhanced HMS formation, in line with our $P_{HMS}$ results. The $P_{HMS}$ estimations also exhibited a good correlation with HMS/CO (R=0.57, P<0.05), and can roughly capture the diurnal variations of HMS/CO during pollution period (**Figure S7a**, **attached below**).

Regarding the comparison between predicted $P_{HMS}$ and HMS measurements, a previous study attempted to correlate $P_{HMS}$ ($\mu g\ m^{-3}\ h^{-1}$) and daily averaged HMS levels by multiplying 24 hours with $P_{HMS}$ as a rough approximation (Shen et al., 2024) given the low reactivity of HMS after its formation although other physical process such as diffusion and deposition can also impact its ambient concentration (Liu et al., 2024). As shown in **Figure R1**, our comparison between predicted HMS and measured HMS levels exhibited a consistent pattern, with higher concentrations during haze events. The averaged HMS level predictions during hazy days were in line with ambient observations, whereas the estimated HMS level was an order of magnitude lower than

the measurements on clean days. This discrepancy can be largely attributed to the fact that ambient HMS levels are influenced by various key atmospheric processes, including precursors emission, physical transport (advection and turbulent diffusion), chemical reactions, and deposition losses, whereas $P_{HMS}$ solely represents the averaged rate for the HMS formation process. While we acknowledge that chemical transport models offer a more comprehensive representation of these atmospheric processes and can provide more precise estimations of HMS levels, the primary focus of our $P_{HMS}$ calculations was to emphasize the enhanced HMS formation during hazy days, as evidenced by the higher HMS/CO ratio, and the significance of aerosol ionic strength in this context. Despite potential uncertainties may persist in $P_{HMS}$ calculations reported here due to the lack of laboratory-based kinetics under higher ionic strength level (i.e., IS > 11 mol kg$^{-1}$) and potential enhancement of HCHO solubility within aerosol water as detailed in the Manuscript (Page 13, Line 35), **these $P_{HMS}$ calculations were intended as an initial approximation to depict the potential fluctuations of HMS formation rates under different pollution conditions, rather than for direct comparison with ambient HMS levels.**

[Figure]

**Figure 2.** The averaged atmospheric characteristics under different pollution scenarios in urban Nanjing (i.e., pre-haze, haze and after) including: HMS and sulfate concentration (c); gas precursors levels (d); aerosol properties such as liquid water content (ALWC), pH and ionic strength (IS) (e); and comparison (f) between HMS/CO ratio and HMS formation rates ($P_{HMS}$).

[Figure]

**Figure S7a.** The comparison between HMS formation rates (P$_{HMS}$) and HMS/CO ratio in urban Nanjing.

[Figure]

**Figure R1.** Comparisons between observed HMS level and predictions using HMS formation rates.

**Comment #5:**

*The HMS to sulfate ratio is mass or molar ratio?*

**Response:** Thanks for the hint. The ratio is molar ratio. We have specified it in the Manuscript and Supporting Information.

**Comment #6:**

*Page 6: Why is an epsilon of 5 for NH3/NH4+ partitioning used for the present study? It is stated that is based on Fairbanks Alaska, but this works investigated environment is significantly different and with higher pollution.*

**Response:** Thanks for the comment. To clarify, we monitored the levels of gas-phase

ammonia during our observations in Nanjing using MARGA to estimate the urban aerosol pH. For marine aerosol, we utilized the $\varepsilon(=NH_4^+/(NH_3+NH_4^+))$ value based on previous observations conducted in the Bohai Sea in May 2021 (Wang et al., 2022), where the $NH_3$ concertation averaged at 3.4±1.6 and the $NH_4^+$ level averaged at 4.0±3.7 µg m$^{-3}$. Considering the resemblance of our observational environment in April 2023 to that of the aforementioned study, here we used a $\varepsilon$ of 0.5 to estimate the marine aerosol pH. We would like to declared that there was an inadvertent citation error in our previous reference and we have corrected this in the Manuscript.

**Page 6, Line 2:** "For marine aerosols, we adopted a particle to particle + gas partitioning fraction of $NH_4^+$ ($\varepsilon$) of 0.5, drawing from prior observations conducted in the Bohai Sea in May (Wang et al., 2022) in light of the absence of gaseous data."

**Comment #7:**

*Page 8 Lines 18-21: This statement is a bit confusing. Since HCHO is a precursor of HMS, wouldn't it be expected to observe a decrease of HCHO and an increase of HMS rather than simultaneous peaks of the two species?*

**Response:** Thanks for the comment. We agree with the reviewer that the formation of HMS entails the consumption of its precursor formaldehyde (HCHO). Nevertheless, given the minimal presence of HMS (specifically, HMS level at 0.36±0.09 µg m$^{-3}$ and HCHO level at 6.72±1.56 µg m$^{-3}$ during hazy days), the generation of HMS likely exerts a negligible influence on ambient HCHO concentrations. In this scenario, it was the level of HCHO that predominantly influences the concentration of HMS.


**Revision in Abstract:**

**Page 2, Line 1:** "Hydroxymethanesulfonate (HMS) has emerged as a critical organosulfur species in ambient aerosols, yet the impact of aerosol properties, particularly ionic strength (IS), on the formation of HMS remains uncertain. Here, HMS levels in wintertime of urban Nanjing, China were quantified at 0.30±0.10 µg m$^{-3}$, where the contribution of in-cloud formation likely carry minor significance due to the barrier resulted from stable stratification."

**Revision in Manuscript:**

**Page 8, Line 1:** "

[revised manuscript text omitted]

**Revision in Supporting Information: "**

[Figure]

**Figure S2.** Ancillary atmospheric measurements in urban Nanjing including fog event criteria based on hourly relative humidity (RH>90%) and Visibility (<1 km) data (e); Time series of the average cloud water content below the planetary boundary layer height over our observation sites (f).

[Figure]

**Figure S3**. Observed vertical temperature profile at 00 UTC during our observation in Nanjing. The dates of hazy days were marked by red.

As for marine environment, we attempt to determine the HMS formation rate in cloud water ($P_{HMS, c}$) and aerosol water ($P_{HMS, a}$) considering the minimal impact of temperature inversion during our observation in April. In the manuscript, we have compared the $P_{HMS, a}$ ($2.06 \times 10^{-5}$ µg m$^{-3}$ h$^{-1}$) in our study with $P_{HMS, c}$ ($1.2 \times 10^{-5}$ µg m$^{-3}$ h$^{-1}$) estimated for observations near Beaufort Sea with a CWC of 4.8 mg m$^{-3}$ and a pH of 5 (Liu et al., 2021a). Using these same CWC and pH, here we calculated the averaged $P_{HMS, c}$ of $4.95 \times 10^{-5}$ µg m$^{-3}$ h$^{-1}$ in our marine environment, which was comparable to the $P_{HMS, a}$. The calculation results and input parameters were summarized in **Table R1**. This comparable $P_{HMS, a}$ and $P_{HMS, c}$ can be explained by that despite that the CWC being ~ 300 times higher than ALWC and aerosol pH being lower, the higher ionic strength level in marine aerosol water can significantly promote HMS formation process, rendering the HMS formation within aerosol water a process comparable to that observed in cloud and fog environments. Altogether, although these formation rates serve as initial approximations, **these results may indicate the significant contribution of both aerosol and cloud water in the formation of HMS in marine atmosphere**. Following discussion has been included in **Section 3.4**.

**Revision in Manuscript:**

**Page 17, Line 10:** "Using the averaged [SO$_{2(g)}$]=0.82 ppb and [HCHO$_{(g)}$]=0.5 ppb (Ervens et al., 2003; Zhao et al., 2024), the potential HMS formation rates in these marine aerosols were calculated to be $2.06 \times 10^{-5}$ µg m$^{-3}$ h$^{-1}$, a value comparable to reported cloud/fog-based HMS production rate ($1.2 \times 10^{-5}$ µg m$^{-3}$ h$^{-1}$) near Beaufort Sea (Liu et al., 2021a) with a liquid water content (LWC) of 4.8 mg m$^{-3}$ and a pH of 5. Considering the minimal impact of temperature inversion during our observation in April, here we also determined the in-cloud HMS formation rate to be $4.95 \times 10^{-5}$ µg m$^{-3}$ h$^{-1}$ in our marine environment using a LWC of 4.8 mg m$^{-3}$ and a pH of 5. These results further highlighted the significance of IS-dependent enhancement in HMS formation, which can render the HMS formation within aerosol water a process comparable to that observed in cloud and fog environments, particularly in humid and pristine conditions."

Besides, for the reviewer's reference, here we also attempt to compared the HMS formation rate in cloud water ($P_{HMS, c}$) and aerosol water ($P_{HMS, a}$) in urban Nanjing although the transportation of in-cloud HMS formation can be largely impeded. Global models (Song et al., 2019) have estimated the cloud pH in the winter of Nanjing aera to be around 4-5, consistent with winter observations in China (4.1±0.6) (Shah et al., 2020). In our analysis, we utilized the upper value of cloud pH of 5 to estimate $P_{HMS, c}$ under pre-haze and haze conditions, comparing it with $P_{HMS, a}$. Despite the tendency for cloud pH to decrease during haze pollution (Li et al., 2017), it was observed that $P_{HMS, c}$ was five times lower than $P_{HMS, a}$ on hazy days. Nevertheless, it is worth noting that the $P_{HMS, a}$ value of 1.06±0.64 × 10$^{-2}$ µg m$^{-3}$ h$^{-1}$ can effectively reflect the HMS levels observed in ambient aerosol, furthering indicate the predominate role of aerosol water for HMS formation during temperature inversion. Besides, the $P_{HMS, c}$ were found to be comparable to $P_{HMS, a}$ on clean days (**Table R1**). Considering possible overestimations in $P_{HMS, c}$ due to reduced precursor concentrations and hindered transport to ground

level resulting from weak vertical exchange, these results suggest the limited impact of in-cloud HMS formation on detected HMS levels. We acknowledge that uncertainties may exist in the calculation of HMS formation rate due to potential variability associated with cloud pH, thus above information was not included in the Manuscript.

**Table R1**. Summary of $P_{HMS, a}$ and $P_{HMS, a}$ as well as input parameters.

| | Marine | | Urban | | | |
|---|---|---|---|---|---|---|
| | **Cloud** | **Aerosol** | **Cloud** | | **Aerosol** | |
| | | | Pre-haze | Haze | Pre-haze | Haze |
| T (K) | 283±1 | 283±1 | 272±2 | 277±4 | 272±2 | 277±4 |
| SO$_2$ (ppb) | 0.82±0.42[a] | 0.82±0.42 | 1.99±0.45 | 2.21±0.69 | 1.99±0.45 | 2.21±0.69 |
| HCHO (ppb) | 0.5[a] | 0.5 | 1.91±1.05 | 5.38±1.08 | 1.91±1.05 | 5.38±1.08 |
| LWC (µg m$^{-3}$) | 4.8[b] mg m$^{-3}$ | 14.72 | 1.5±2.2 mg m$^{-3}$ | 5.12±5.45 mg m$^{-3}$ | 22.69±20.65 | 80.45±38.40 |
| pH | 4.8[b] | 4.24±0.15 | 5[d] | 5 | 5.25±0.42 | 4.76±0.46 |
| Ionic strength (mol kg$^{-1}$) | 10$^{-4}$ [c] | 4.30±1.24 | 10$^{-4}$ | 10$^{-4}$ | 12.32±3.19 | 8.85±1.30 |
| P$_{HMS}$ ($\times$ 10$^{-4}$ µg m$^{-3}$ h$^{-1}$) | 0.51 | 0.21 | 2.56±4.68 | 20.4±23.3 | 6.40±6.96 | 106±64.6 |

[a] The levels of SO$_2$ and HCHO over marine environments were adopted from previous studies (Ervens et al., 2003; Zhao et al., 2024).
[b] The LWC and pH for cloud/fog droplets came from previous coastal study near Beaufort Sea (Liu et al., 2021a).
[c] This typical ionic strength value of cloud droplet was given by Herrmann et al. (2015).
[d] This cloud PH was the upper value of global estimation on Nanjing aera (Song et al., 2019).

**Major Comment #2:**

*2.1 Section 3.3 lacks clarity. In Page 10 Line 41, the authors suggest a correlation between HMS and ALWC. This should not be surprising, as sulfate is highly hygroscopic. This correlation could simply reflect the high correlation between HMS and sulfate (Page 8, Line 26).*

**Response:** We agree with the reviewer that sulfate ions can contribute to the ALWC given its hygroscopicity. However, sulfate ions' share in the total water-soluble inorganic ions was relatively minor, averaging at 13±4%. Here, we have also estimated the ALWC without factoring in sulfate ions (referred to as ALWC-non-sulfate). The ALWC-non-sulfate values were generally lower but exhibited a strong correlation with ALWC (R=0.98, P<0.01), as illustrated in **Figure R1**. Additionally, HMS levels also displayed a notable correlation with ALWC-non-sulfate (R=0.67, P<0.01). In conjunction with our response to **Major Comment #1**, where we discussed the likely insignificant contribution of in-cloud HMS formation to the observed HMS levels in urban Nanjing, here the strong HMS-ALWC correlation may indicate that aerosol water

can serve as an important medium for HMS formation.

[Figure]

**Figure R1**. (a) Time series of HMS and sulfate level during the winter of Nanjing; (b) estimated aerosol liquid water content with and without the consideration of sulfate ions.

*2.2 For Figure 3, it is expected that ionic strength inversely correlates with RH due to dilution effects at high humidity (high RH leads to high ALWC and therefore diluted solutions). Figure 3 does not make a good argument for ionic strength as a controlling factor in HMS formation. The right panel relies entirely on model output without supporting observational data, making the conclusion speculative. There is simply no direct evidence that ionic strength itself is influencing HMS formation. The observed increase in HMS during hazy days could stem from several other factors: (1) Increased ALWC (Figure 2e), which enhances the dissolution of $SO_2$ and HCHO, potentially accelerating HMS production. (2) Elevated HCHO (Figure 2d), which increases by a factor of three during haze events and could drive more HMS formation. These alternative explanations are not adequately addressed in the manuscript.*

**Response:** Thanks for the comment. We would like to declare that **Figure 3** was designed to declare the inverse relationship between ionic strength (IS) and RH (**Figure 3a**) and the discontinuous IS-dependent enhancement on HMS formation (**Figure 3b**), rather than to highlight the controlling role of ionic strength in HMS formation. We also attached **Figure 3** below from our manuscript for the reviewer's reference. Regarding the impact of IS on HMS formation, we have calculated the $P_{HMS}$ without considering the IS. The results shows that even with a 4-fold increase in ALWC coupled with a 2-fold increase in HCHO levels, these factors cannot completely counterbalance the potential 10-fold reduction in HMS formation rates resulting from the decreased aerosol pH (**Tabel S1**, attached below), resulting in slower $P_{HMS}$ under hazy days ($1.2\pm1.7\times 10^{-4}$ $\mu g$ $m^{-3}$ $h^{-1}$) compared to clean periods ($3.2\pm5.4\times 10^{-4}$ $\mu g$ $m^{-3}$ $h^{-1}$) as shown in **Figure S9c**. Here we also quantified the relative variations in HMS formation rate during hazy days compared to clean days ($\Delta P/P_{HMS, \text{clean}}$) corresponding to changes in $SO_2$ level,

HCHO level, temperature, ALWC, aerosol pH and ionic strength. The results were added in Supporting Information as **Figure S9d**. And it can be seen that it was the reduction in aerosol ionic strength on polluted days which exhibited more pronounced enhancement in HMS formation, ultimately leading to a nearly 10-fold rise in $P_{HMS}$ during haze episode compared to dry and clean days (**Tabel S1**). Following edits have been made in the Manuscript and Supporting Information.

**Page 13, Line 0:** "The daily averaged steady-state $P_{HMS}$ throughout the sampling period in Nanjing was further determined (as detailed in **Section S5**) to explore the potential role of aerosol properties such as acidity and ionic strength in ambient HMS formation. Given that carbon monoxide (CO) was usually considered to be an inert chemical species during rapid haze formation, with its increase often interpreted as indicative of the accumulation of primary pollutants in the shallower boundary layer (Williams et al., 2016), the ratio of HMS to CO (HMS/CO) was calculated to better represent the secondary formation of ambient HMS (**Figure 2f**). During our study period, enhanced HMS formation was noted during hazy days as evident by the higher HMS/CO ratio (0.51±0.11). The $P_{HMS}$ estimations exhibited a good correlation with HMS/CO (R=0.57) (**Figure S9a**), and can roughly capture the diurnal variations of HMS/CO during pollution period. Besides, elevated HMS formation rates were found during haze event with averaged $P_{HMS}$ of $5.8\pm5.9 \times 10^{-3}$ μg m$^{-3}$ h$^{-1}$, an order of magnitude higher than that during the clean periods ($4.5\pm8.4\times 10^{-4}$ μg m$^{-3}$ h$^{-1}$) (**Figure 2f**). Certainly, when the ionic strength effect in aerosol water was not considered, daily $P_{HMS}$ was generally calculated to be one to two orders of magnitude lower (**Figure S9b**). Noteworthy, even with a 4-fold increase in ALWC coupled with a 2-fold increase in HCHO levels, these factors cannot completely counterbalance the potential 10-fold reduction in HMS formation rates resulting from the decreased aerosol pH, leading to slower HMS formation rates ($1.2\pm1.7\times 10^{-4}$ μg m$^{-3}$ h$^{-1}$) during hazy days compared to clean days ($3.3\pm5.3\times 10^{-4}$ μg m$^{-3}$ h$^{-1}$) (**Figure S9c**), failing to explain the higher HMS/CO ratio and HMS levels. Similar discrepancies have also been highlighted in previous studies where estimated HMS formation rates, without the consideration of high IS level in aerosol water, inadequately represented the ambient HMS concentrations and their temporal fluctuations (Ma et al., 2020; Campbell et al., 2022; Zhao et al., 2024). For instance, prior study has reported a $P_{HMS}$ of $2.6 \times 10^{-2}$ μg m$^{-3}$ h$^{-1}$, which was estimated using parameters (i.e., SO$_2$ solubility, $k_1$ and $k_2$) obtained in dilute solution and thus failed to represent the significant HMS level in Beijing (up to 18.5 μg m$^{-3}$) (Ma et al., 2020). These results raised a possibility that the HMS formation rate could be largely underestimated without considering high ionic strength level in aerosol water. During our observation, we found that it was the reduction in aerosol ionic strength on humid and polluted days led to more pronounced enhancement in HMS formation, This ultimately led to a nearly 10-fold rise in $P_{HMS}$ during haze episode compared to dry and clean days (**Figure S9d**), contributing to the elevated HMS level and HMS/CO ratio during haze event."

[Figure]

**Figure 3**. (a) The correlation between aerosol ionic strength and relative humidity level; (b) The enhancement factor (EF) exerts by ionic strength on the HMS formation rate, calculated as $EF=P_{HMS}/P_{HMS, dilute}$ at pH of 4. $P_{HMS, dilute}$ was calculated using the parameter proposed for dilute cloud water when the ionic strength was smaller than 1 mol kg$^{-1}$. Beyond this threshold (IS≥1 mol kg$^{-1}$), the $P_{HMS}$ was calculated following the equations given in **Table S3**.

[Figure]

**Figure S9.** (a) The comparison between estimated HMS formation rate ($P_{HMS}$) with HMS/CO ratio in urban Nanjing; (b) The comparison between HMS/CO ratio and $P_{HMS}$ without considering the IS effect in urban Nanjing; (c) the averaged $P_{HMS}$ before, within and after haze event, with and without consideration of ionic strength. The formation

rate estimations on December 19$^{th}$, 2023 and January 3$^{rd}$, 2024 are not feasible due to the absence of HCHO level and aerosol properties, respectively; (d) The relative variation in HMS formation rate during hazy days ($\Delta P/P_{HMS, \, clean}$) corresponding to changes in SO$_2$ level, HCHO level, temperature, ALWC, pH and ionic strength compared to clean days."

**Table S1.** Summary of atmospheric measurements in this study.

| | Continental aerosol | | | Marine aerosol |
|---|---|---|---|---|
| | **Pre-haze** | **Haze event** | **After haze** | |
| T (K) | 272±2 [a] | 277±4 | 277±1 | 283±1 |
| RH | 52±17% | 74±8% | 69±5% | 83±6% |
| PM$_{2.5}$ (µg m$^{-3}$) | 38.65±15.66 | 114.29±18.01 | 73.27±33.32 | - [b] |
| SO$_2$ (ppb) | 1.99±0.45 | 2.21±0.69 | 2.18±0.54 | 0.82±0.42 |
| HCHO (ppb) | 1.91±1.05 | 5.38 ±1.08 | 3.67±0.68 | - |
| HMS (µg m$^{-3}$) | 0.23±0.08 | 0.36±0.09 | 0.32±0.08 | 0.050±0.012 |
| Sulfate (µg m$^{-3}$) | 4.18±1.71 | 11.44±4.07 | 8.87±4.05 | 2.09±0.46 |
| HMS/Sulfate (%) | 5.87±1.70% | 3.36±0.73% | 4.15±1.40% | 2.57±0.09% |
| ALWC (µg m$^{-3}$) | 22.69±20.65 | 80.45±38.40 | 61.30±36.34 | 14.72±6.40 |
| pH | 5.25±0.42 | 4.76±0.46 | 3.95±0.53 | 4.24±0.15 |
| Ionic strength (mol kg$^{-1}$) | 12.32±3.19 | 8.85±1.30 | 10.39±0.85 | 4.30±1.24 |
| P$_{HMS}$ ($\times 10^{-4}$ µg m$^{-3}$ h$^{-1}$) | 4.54±8.4 | 57.6±58.7 | 1.18±1.87 | 0.26 [c] |
| HMS/CO | 0.38±0.14 | 0.51±0.11 | 0.35±0.12 | - |

[a] The numerical representation of average ± one standard deviation.

[b] The data was not available.

[c] The formation rate was calculated using averaged values listed above assuming the HCHO level of 0.5 ppb (Wagner et al., 2001; Anderson et al., 2017).

**Major comment #3:**

*3. Ionic Strength and pH Calculations. The calculation of ionic strength lacks details. Although ISORROPIA II can calculate ionic strength, the authors use the AIOMFAC model without discussing its added value. In particular, how important are organic species to ionic strength and pH?*

**Response:** In this work, we actually integrated the ISORROPIA II and AIOMFAC model to estimate ALWC, pH, and ionic strength (IS) as detailed in **Section S4**. This method has been proposed and validated by previous study (Battaglia et al., 2019). Briefly, for urban aerosols, inorganic PM composition (SO$_4^{2-}$, NO$_3^-$, Cl$^-$, Na$^+$, NH$_4^+$, Ca$^{2+}$, K$^+$ and Mg$^{2+}$) and gaseous HNO$_3$, NH$_3$, HCl data obtained by MARGA together

with temperature and RH, were input into ISORROPI II under "Forward" mode and "metastable" state to derive equilibrium concentrations of ALWC and all ionic species. For marine aerosols, we adopted a particle to particle + gas partitioning fraction of $NH_4^+$ ($\varepsilon$) of 0.5, drawing from prior observations conducted in the Bohai Sea in May (Wang et al., 2022a) in light of the absence of gaseous data. Subsequently, organic components in conjunction with the inorganic matrix outputs from ISORROPIA II were incorporated into AIOMFAC model at the same temperature. During AIOMFAC run, it was required the inorganic species inputs entered as ionic pairs to ensure electroneutrality. As suggested by Battaglia et al. (2019), the ionic pairs were assigned in a way detailed in the Supporting Information (**Section S4**). When introducing the organics into AIOMFAC, all species (both inorganic and organic) were inputted in mole fractions. In this study, the mole contribution of organics was calculated based on the measured WSOC mass concentration. And all the WSOC were assumed to be nonacid given the minimal impact of organic acids on aerosol properties (Song et al., 2018; Battaglia et al., 2019; Nah et al., 2019). Considering the typical hygroscopicity (k) ranging from 0.1 to 0.2 and the composition of organic aerosols (Liu et al., 2021b; Pöhlker et al., 2023), levoglucosan ($C_6H_{10}O_5$, k=0.16±0.01) (Petters and Kreidenweis, 2007) were selected as model organic species found in ambient aerosols. **The model then assumed that water made up the difference between the sum of mole fractions of all organic and inorganic inputs and unity, with the water activity ($a_w$) equating to the ambient relative humidity (RH).** Then, the total moles of inputs were adjusted manually while maintaining the relative ratios between each species constant until the RH values predicted by AIOMFAC were within 1% error of the RH value set for the ISORROPIA II model. The output of AIOMFAC was used to determine the ALWC, pH, and ionic strength. A flow diagram of above processes was prepared and added into the Supporting Information as **Figure S14**. It was acknowledged that the model outputs may not completely represent the hygroscopic nature of organic aerosols because of incomplete information concerning the speciation and concentrations of organic components. Furthermore, the model did not account for the re-equilibration of gas-phase species with the altered water content caused by the organic species through AIOMFAC, leading to estimation bias in ALWC. Above information has been provided in the Supporting Information.

[Figure]

**Figure S14**. Flow diagram for incorporating ISORROPIA II and AIOMFAC to estimate the aerosol properties with consideration of organics. Total moles faction of the organic–inorganic mixed system inputs were adjusted but keep their relative ratio constant until the AIOMFAC-output RH was within 1% of the ambient RH level. Given that the difference between the total inputs and unity was contributed by aerosol water, the aerosol liquid water content together with aerosol pH and ionic strength can be determined according to **Eqn. S2** and **Eqn. S3**.

The impact of organics in determining the aerosol properties were discussed in **Section S4** by comparing our estimations with inorganic-only simulations from ISORROPIA II. Here we also attached **Figure S15** for the reviewer's reference. Briefly, there was consistent patterns in aerosol properties variations across different pollution conditions regardless of the inclusion of organic aerosols with higher ALWC and lower pH and ionic strength estimations for humid and polluted days as summarized in **Table R2**. When organic components were considered, ALWC values were estimated to be approximately 20±8% higher during hazy days and 48±10% higher on clean days compared to inorganic-only estimations. The more significant variance on clean days was attributed to a greater contribution of water-soluble organics to $PM_{2.5}$ (20±14%) compared to hazy days (10±2%). Additionally, the inclusion of organics led to slightly elevated aerosol pH values ($\Delta pH \leq 0.5$ unit), in line with prior research (Battaglia et al., 2019). This slight pH increase can be attributed to non-dissociating organic compounds in aerosols enhancing water absorption and subsequently reducing $H^+$ concentration. These minor deviations hint at the predominant influence of inorganic constituents on aerosol pH levels. With the consideration of organics, this work estimated lower ionic strengths (**Table R2**) during our observation in Nanjing, with more substantial reduction observed on dry and clean days (48±33%) compared to humid and polluted period (25±17%). The pronounced differences in ionic strength estimations under dry and clean conditions can be partially explained by the increase in ALWC (up to 50%), driven by the hygroscopic nature of organic species as the rise in ALWC resulted in significant dilution of the aerosol water, leading to reduced concentrations of ionic species. A previous study utilizing ISORROPIA II also noted higher ionic strength under low humidity conditions that the ionic strength could exceed 60 mol kg$^{-1}$ under 40% RH and drop sharply to less than 10 mol kg$^{-1}$ when RH increased over 80% with constant aerosol composition (Song et al., 2018). Besides, a measurement-based study reported a much narrower range of ionic strength for urban aerosols in Los Angeles (ranging from 8.0 to 18.6 mol kg$^{-1}$) based on measured ionic concentration and aerosol water mass content (Stelson and Seinfeld, 1981) where aerosol mass concentration ranged from 82 to 192 µg m$^{-3}$ with RH of ~ 55%. These results suggested the important role of organic species in determining aerosol ionic strength, primarily by impacting the aerosol water content. Overall, it was anticipated that the inclusion of organic components in thermodynamic models can lead to more precise representations of aerosol properties, particularly under lower humidity conditions. However, we acknowledge that potential uncertainties may persist in our estimations, particularly

under lower humidity, as ambient aerosols containing water-soluble inorganic and organic components may undergo phase separation, resulting in uneven phase distribution and mixing states. Additionally, we compared the HMS formation rates ($P_{HMS}$) using the aerosol properties estimation with and without consideration of impact of organics (**Figure S15d**). The results revealed that haze events consistently exhibited higher formation rates compared to clean days, regardless of the inclusion of organics and $P_{HMS}$ for clean days under inorganic-only scenarios were notably lower due to lower ALWC and pH levels, alongside higher ionic strength. Above information has been also added into **Section S4**.

**Page 16, Line 32**: "The model output results have been discussed in manuscript and its comparison with inorganic-only simulations from ISORROPIA II for urban aerosols was shown in **Figure S15**. Briefly, there was consistent patterns in aerosol properties variations across different pollution conditions regardless of the inclusion of organic aerosols with higher ALWC and lower pH and ionic strength estimations for humid and polluted days. When organic components were considered, ALWC values were estimated to be approximately 20±8% higher during hazy days and 48±10% higher on clean days compared to inorganic-only estimations. The more significant variance on clean days was attributed to a greater contribution of water-soluble organics to $PM_{2.5}$ (20±14%) compared to hazy days (10±2%). Additionally, the inclusion of organics led to slightly elevated aerosol pH values ($\Delta pH \leq 0.5$ unit), in line with prior research (Battaglia et al., 2019). This slight pH increase can be attributed to non-dissociating organic compounds in aerosols enhancing water absorption and subsequently reducing $H^+$ concentration. These minor deviations hint at the predominant influence of inorganic constituents on aerosol pH levels. With the consideration of organics, this work estimated lower ionic strengths during our observation in Nanjing, with more substantial reduction observed on dry and clean days (48±33%) compared to humid and polluted period (25±17%). The pronounced differences in ionic strength estimations under dry and clean conditions can be partially explained by the increase in ALWC (up to 50%), driven by the hygroscopic nature of organic species. This rise in ALWC resulted in significant dilution of the aerosol solution, leading to reduced concentrations of ionic species. A previous study utilizing ISORROPIA II also noted higher ionic strength under low humidity conditions that the ionic strength could exceed 60 mol kg$^{-1}$ under 40% RH and drop sharply to less than 10 mol kg$^{-1}$ when RH increased over 80% with constant aerosol composition (Song et al., 2018). Besides, a measurement-based study reported a much narrower range of ionic strength for urban aerosols in Los Angeles (ranging from 8.0 to 18.6 mol kg$^{-1}$) based on measured ionic concentration and aerosol water mass content (Stelson et al., 1981) where aerosol mass concentration ranged from 82 to 192 μg m$^{-3}$ with RH of ~ 55%. These results suggested the important role of organic species in determining aerosol ionic strength, primarily by impacting the aerosol water content. Overall, it was anticipated that the inclusion of organic components in thermodynamic models can lead to more precise representations of aerosol properties, particularly under lower humidity conditions. However, it is important to acknowledge potential uncertainties in our estimations, particularly under lower humidity, as ambient aerosols containing water-soluble inorganic and organic

components may undergo phase separation, resulting in uneven phase distribution and mixing states. Additionally, we compared the HMS formation rates ($P_{HMS}$) using the aerosol properties estimation with and without consideration of impact of organics (**Figure S15d**). The results revealed that haze events consistently exhibited higher formation rates compared to clean days, regardless of the inclusion of organics and $P_{HMS}$ for clean days under inorganic-only scenarios were notably lower due to lower ALWC and pH levels, alongside higher ionic strength."

[Figure]

**Figure S15**. The comparison between estimated aerosol properties (a-c) including (a) aerosol liquid water content (ALWC); (b) aerosol pH; (c) aerosol ionic strength; and (d) $P_{HMS}$ with and without the consideration of organic aerosols.

**Table R2**. Comparison in aerosol properties and $P_{HMS}$ with and without the consideration of organics.

| | | Pre-haze | Haze event | After haze |
|---|---|---|---|---|
| | RH | 52±17% | 74±8% | 69±5% |
| Inorganic-only estimation | ALWC ($\mu g\ m^{-3}$) | 15.2±18.8 | 64.6±35.9 | 41.8±25.4 |
| | pH | 5.1±0.4 | 4.1±0.4 | 3.6±0.7 |
| | Ionic strength (mol kg$^{-1}$) | 25.1±11.9 | 11.9±1.1 | 17.5±2.6 |
| | $P_{HMS}$ ($\mu g\ m^{-3}\ h^{-1}$) | 4.8±10.7×10$^{-9}$ | 1.0±3.3×10$^{-2}$ | 0.8±0.3×10$^{-6}$ |

| Inorganic + Organics estimations | ALWC ($\mu g\ m^{-3}$) | 22.7±20.65 | 84.5±38.4 | 61.3±36.4 |
|---|---|---|---|---|
| | pH | 5.3±0.4 | 4.7±0.5 | 3.9±0.5 |
| | Ionic strength (mol kg$^{-1}$) | 12.3±3.2 | 8.8±1.3 | 10.4±0.8 |
| | $P_{HMS}$ ($\mu g\ m^{-3}\ h^{-1}$) | 4.54±8.4×10$^{-4}$ | 57.6±58.7×10$^{-4}$ | 1.18±1.87×10$^{-4}$ |

**Major Comment #4:**

*4. The pH calculation (Page 5, Line 39) references the use of gas-phase species such as NH3 and HNO3, but these concentrations are not shown. Providing this information would help understand the pH and ionic strength estimates.*

**Response:** Thanks for the advice. The daily averaged concentration of NH$_3$ and HNO$_3$ were added into Supporting information as **Figure S2c**.

[Figure]

**Figure S2.** Ancillary atmospheric measurements in urban Nanjing including gas-phase NH$_3$ and HNO$_3$ level (c).

**Minor Comment #5:**

*5. Figure 1. The authors might consider adding panel titles to clearly distinguish between two different measurements (left vs. right panels).*

**Response:** We appreciate the suggestion. Panel titles have been included in **Figure 1**. Please kindly refer to our updates below.

[Figure]

**Figure 1.** Left panel (Urban Observations): Time series of atmospheric measurements in urban Nanjing, including temperature and relative humidity (a); $SO_2$ and HCHO levels (b); HMS and sulfate concentration (c); $PM_{2.5}$ composition (d). The gray shade area highlighted a 7-days' haze episode (December 27th, 2023 to January 2nd, 2024) with $PM_{2.5}$ mass concentration exceeding 75 μg m$^{-3}$. Other ancillary measurements in urban atmosphere including wind speed and wind direction, $PM_{2.5}$ mass concentration, $O_3$, $NH_3$ level, fog occurrence and cloud water content can be found in **Figure S2**. Right panel (Marine Observations): Atmospheric measurements during marine cruise, including temperature and relative humidity (e); $SO_2$ (f), HMS and sulfate level where the biogenic nss-sulfate (bio-nss-sulfate) was calculated based on previously measured bio-nss-sulfate/nss-sulfate ratio of 0.14 (Yang et al., 2015) (g); $PM_{2.5}$ composition (h).

---

## Author Response (AR2)

Dear Editor,

We are grateful for receiving your thoughtful feedback. And we thank for the time and effort invested by both the editor and the reviewers in providing constructive suggestions that have significantly contributed to enhancing the quality of this work, particularly in addressing the role of in-cloud HMS formation during our observations, a point highlighted by all three reviewers. We acknowledge that despite the acceptance of our explanations and revisions by two reviewers, Anonymous Referee #3 may still have consideration on this point, alongside other comments. Therefore, please find below our detailed point-to-point responses to the comments from Anonymous Referee #3. The referee's comments are presented in italics, followed by our responses in normal font.

***Comment # 1***

*Given the short lifetime of clouds (approximately 20-30 minutes), it is inherently difficult to rule out the possibility of cloud processing even on days classified as "clear." Therefore, the evidence for HMS formation occurring solely in aerosols under urban conditions-without any contribution from cloud chemistry-remains limited.*

**Response:** Thanks for the comment. We agree with the reviewer that the contribution of in-cloud HMS formation cannot be completely ruled out during our observations in urban Nanjing. However, as indicated in our previous response, the prevalent presence of temperature inversions (**Figure S3**, attached below) across our urban observations could largely impede the transportation of gas precursors emitted near surface to high altitudes and the descent of chemicals produced in high-altitude clouds to ground level (Wang et al., 2022). Besides, low wind speeds (**Figure S2a**) were observed during our observations. These conditions may suggest that even if HMS were largely formed with cloud droplets, its contribution to near surface aerosol were largely inhibited. Furthermore, days lacking temperature inversions (from December 19$^{th}$ to December 23$^{rd}$, 2023) were associated with negligible cloud water content (**Figure S2f**), which were obtained from MERRA-2 (Modern-Era Retrospective analysis for Research and Applications, Version 2) (Gelaro et al., 2017). And the lower levels of HMS observed in this work compared to other polluted areas such as Northern China (Ma et al., 2020; Wei et al., 2020; Wang et al., 2024) may also suggest the minor contribution from in-cloud formation.

In addition, we attempted to compare the HMS formation rate in cloud water ($P_{HMS, c}$) and aerosol water ($P_{HMS, a}$) in urban Nanjing, despite the potential hindrance to in-cloud HMS formation transport. Global models (Song et al., 2019) have estimated the cloud pH in the winter of Nanjing aera to be around 4-5, consistent with winter observations in China (4.1±0.6) (Shah et al., 2020). In our analysis, we utilized the upper value of cloud pH of 5 to estimate $P_{HMS, c}$ under pre-haze and haze conditions, comparing it with $P_{HMS, a}$ (**Table R1**). Despite the tendency for cloud pH to decrease

during haze pollution (Li et al., 2017), it was observed that $P_{HMS, c}$ was 2-5 times lower than $P_{HMS, a}$ in the winter of Nanjing. Nevertheless, it is worth noting that the $P_{HMS, a}$ value of $1.06\pm0.64 \times 10^{-2}\,\mu g\,m^{-3}\,h^{-1}$ during hazy days can effectively reflect the HMS levels observed in ambient aerosol, furthering indicate the predominate role of aerosol water for HMS formation during temperature inversion. We acknowledged that uncertainties may exist in the calculation of $P_{HMS, c}$ due to reduced precursor concentrations resulting from weak vertical exchange and potential variability associated with cloud pH, thus above information was not included in the Manuscript.

Overall, **while we recognize that the HMS formation in cloud layers characterized by persistent dense clouds and extended durations may carry great contributions, in the context of this study, the presence of temperature inversions, low wind speeds, and diminished cloud water content during the winter season in Nanjing suggests that the impact of in-cloud HMS formation on the observed levels of particulate HMS may be insignificant**. Further discussion on the role of in-cloud HMS chemistry was expanded in the Manuscript.

**Revision in Manuscript:**

**Page 19, Line 29:** "Collectively, this study provided valuable information for the prevalence of HMS and the validation the model-derived outcomes concerning HMS quantification. The work primarily concentrated on particulate HMS formation in aerosol liquid water, highlighting the role of moderate-level ionic strength in atmospheric HMS formation, advocating for their integration into global or regional models to better represent the particulate sulfur chemistry, especially in humid environments. Nevertheless, it is noted that in-cloud HMS chemistry may also contribute to the particulate HMS levels where vertical and high-altitude observations are required to fully understand its significance, thus warranting further investigation."

[Figure]

**Figure S2.** Ancillary atmospheric measurements in urban Nanjing including wind speed and wind direction (a); Time series of the average cloud water content below the planetary boundary layer height over our observation sites (f). Gray shadow indicates the haze period.

[Figure]

**Figure S3**. Observed vertical temperature profile at 00 UTC during our observation in Nanjing. The dates of hazy days were marked by red.

**Table R1**. Summary of $P_{HMS, a}$ and $P_{HMS, a}$ as well as input parameters.

| | Urban Nanjing | | | |
|---|---|---|---|---|
| | **Cloud** | | **Aerosol** | |
| | Pre-haze | Haze | Pre-haze | Haze |
| **T (K)** | 272±2 | 277±4 | 272±2 | 277±4 |
| **SO₂ (ppb)** | 1.99±0.45 | 2.21±0.69 | 1.99±0.45 | 2.21±0.69 |
| **HCHO (ppb)** | 1.91±1.05 | 5.38 ±1.08 | 1.91±1.05 | 5.38 ±1.08 |
| **LWC (μg m⁻³)** | 1.5±2.2 mg m⁻³ | 5.12±5.45 mg m⁻³ | 22.69±20.65 | 80.45±38.40 |
| **pH** | 5 [a] | 5 | 5.25±0.42 | 4.76±0.46 |
| **Ionic strength (mol kg⁻¹)** | 10⁻⁴ [b] | 10⁻⁴ | 12.32±3.19 | 8.85±1.30 |
| **P_HMS (× 10⁻⁴ μg m⁻³ h⁻¹)** | 2.56±4.68 | 20.4±23.3 | 6.40±6.96 | 106±64.6 |

[a] This cloud pH was the upper value of global estimation on Nanjing aera (Song et al., 2019).
[b] This typical ionic strength value of cloud droplet was given by Herrmann et al. (2015).

*Comment # 2*

*The proposed role of ionic strength in HMS formation is largely based on model calculations. In the current version of the paper, there is no direct observational evidence presented to support the effect of ionic strength.*

**Response:** Thanks for the feedback. It appears that the "model calculation" mentioned likely pertains to our calculated HMS formation rates ($P_{HMS}$). In this work, these calculated $P_{HMS}$ were compared with observed HMS to carbon dioxide (CO) ratio to highlight the potential role of ionic strength in HMS formation, as elaborated in the **Manuscript (Page 13)**. Initially, considering the low reactivity of HMS, we utilized the ratio of HMS to CO observations (HMS/CO) to better represent the secondary formation of ambient HMS as CO was usually considered to be an inert chemical species during rapid haze formation, with its variation often interpreted as indicative of the diffusion or accumulatio of primary pollutants in the shallower boundary layer (Williams et al., 2016). During our study period, enhanced HMS formation was noted during hazy days as evident by the higher HMS/CO ratio (0.51±0.11) (**Figure 2f,** attached below). Additionally, we compared the $P_{HMS}$ calculations with and without the consideration of ionic strength. With the consideration of ionic strength, $P_{HMS}$ calculations exhibited a good correlation with HMS/CO (R=0.57) (**Figure S9a**). More importantly, we identified increased HMS formation rates during haze events, with average $P_{HMS}$ of $5.8\pm5.9 \times 10^{-3}$ µg m$^{-3}$ h$^{-1}$ compared to that during the clean periods ($4.5\pm8.4\times 10^{-4}$ µg m$^{-3}$ h$^{-1}$), aligning well with the HMS/CO ratio (**Figure 2f**). Noteworthy, when the ionic strength effect in aerosol water was not considered, daily $P_{HMS}$ was generally calculated to be one to two orders of magnitude lower (**Figure S9b**) and slower $P_{HMS}$ were determined under hazy days ($1.2\pm1.7\times 10^{-4}$ µg m$^{-3}$ h$^{-1}$) compared to clean periods ($3.2\pm5.4\times 10^{-4}$ µg m$^{-3}$ h$^{-1}$) (**Figure S9c**), failing to explain the observed higher HMS/CO ratio. We have also quantified the relative variations in HMS formation rate during hazy days compared to clean days ($\Delta P/P_{HMS, clean}$) corresponding to changes in $SO_2$ level, HCHO level, temperature, aerosol liquid water content (ALWC), aerosol pH and ionic strength. And it can be seen that even with a 4-fold increase in ALWC coupled with a 2-fold increase in HCHO levels (**Table S1**, attached below), these factors cannot completely counterbalance the potential 10-fold reduction in HMS formation rates resulting from the decreased aerosol pH and it was the reduction in aerosol ionic strength on polluted days which exhibited more pronounced enhancement in HMS formation, ultimately leading to a nearly 10-fold rise in $P_{HMS}$ during haze episode compared to dry and clean days (**Figure S9d**). Taken together, these findings underscore the pivotal role of aerosol ionic strength in influencing HMS formation in our study.

[Figure]

**Figure 2f.** Comparison between HMS/CO ratio and HMS formation rates ($P_{HMS}$).

[Figure]

**Figure S9.** (a) The comparison between estimated HMS formation rate ($P_{HMS}$) with HMS/CO ratio in urban Nanjing; (b) The comparison between HMS/CO ratio and $P_{HMS}$ without considering the IS effect in urban Nanjing; (c) the averaged $P_{HMS}$ before, within and after haze event, with and without consideration of ionic strength. The formation rate estimations on December 19th, 2023 and January 3rd, 2024 are not feasible due to the absence of HCHO level and aerosol properties, respectively; (d) The relative variation in HMS formation rate during hazy days ($\Delta P/P_{HMS, clean}$) corresponding to changes in $SO_2$ level, HCHO level, temperature, ALWC, pH and ionic strength compared to clean days.

**Table S1.** Summary of atmospheric measurements in this study.

| | Continental aerosol | | | Marine aerosol |
|---|---|---|---|---|
| | **Pre-haze** | **Haze event** | **After haze** | |
| T (K) | 272±2 [a] | 277±4 | 277±1 | 283±1 |
| RH | 52±17% | 74±8% | 69±5% | 83±6% |
| PM$_{2.5}$ ($\mu$g m$^{-3}$) | 38.65±15.66 | 114.29±18.01 | 73.27±33.32 | - [b] |
| SO$_2$ (ppb) | 1.99±0.45 | 2.21±0.69 | 2.18±0.54 | 0.82±0.42 |
| HCHO (ppb) | 1.91±1.05 | 5.38 ±1.08 | 3.67±0.68 | - |
| HMS ($\mu$g m$^{-3}$) | 0.23±0.08 | 0.36±0.09 | 0.32±0.08 | 0.050±0.012 |
| Sulfate ($\mu$g m$^{-3}$) | 4.18±1.71 | 11.44±4.07 | 8.87±4.05 | 2.09±0.46 |
| HMS/Sulfate (%) | 5.87±1.70% | 3.36±0.73% | 4.15±1.40% | 2.57±0.09% |
| ALWC ($\mu$g m$^{-3}$) | 22.69±20.65 | 80.45±38.40 | 61.30±36.34 | 14.72±6.40 |
| pH | 5.25±0.42 | 4.76±0.46 | 3.95±0.53 | 4.24±0.15 |
| Ionic strength (mol kg$^{-1}$) | 12.32±3.19 | 8.85±1.30 | 10.39±0.85 | 4.30±1.24 |
| P$_{HMS}$ ($\times$ 10$^{-4}$ $\mu$g m$^{-3}$ h$^{-1}$) | 4.54±8.4 | 57.6±58.7 | 1.18±1.87 | 0.26 [c] |
| HMS/CO | 0.38±0.14 | 0.51±0.11 | 0.35±0.12 | - |

[a] The numerical representation of average ± one standard deviation.

[b] The data was not available.

[c] The formation rate was calculated using averaged values listed above assuming the HCHO level of 0.5 ppb (Wagner et al., 2001; Anderson et al., 2017).

In addition, we have revised the caption of **Figure S1** in accordance with the guidance provided by the editorial office. We would like to express our gratitude to the editor again for your attention to our work!

Best regards,

Rongshuang XU

School of Ecology and Applied Meteorology,

Nanjing University of Information Science & Technology

Email: rongs_xu@nuist.edu.cn

**References:**

Anderson, D. C., Nicely, J. M., Wolfe, G. M., Hanisco, T. F., Salawitch, R. J., Canty, T. P., Dickerson, R. R., Apel, E. C., Baidar, S., Bannan, T. J., Blake, N. J., Chen, D., Dix, B., Fernandez, R. P., Hall, S. R., Hornbrook, R. S., Gregory Huey, L., Josse, B., Jöckel, P., Kinnison, D. E., Koenig, T. K., Le Breton, M., Marécal, V., Morgenstern, O., Oman, L. D., Pan, L. L., Percival, C., Plummer, D., Revell, L. E., Rozanov, E., Saiz-Lopez, A., Stenke, A., Sudo, K., Tilmes, S., Ullmann, K., Volkamer, R., Weinheimer, A. J. and Zeng, G.: Formaldehyde in the Tropical Western Pacific: Chemical Sources and Sinks, Convective Transport, and Representation in CAM-Chem and the CCMI Models, Journal of Geophysical Research: Atmospheres, **122**(20): 11,201-211,226, https://doi.org/10.1002/2016JD026121, 2017.

Gelaro, R., McCarty, W., Suárez, M. J., Todling, R., Molod, A., Takacs, L., Randles, C., Darmenov, A., Bosilovich, M. G., Reichle, R., Wargan, K., Coy, L., Cullather, R., Draper, C., Akella, S., Buchard, V., Conaty, A., da Silva, A., Gu, W., Kim, G. K., Koster, R., Lucchesi, R., Merkova, D., Nielsen, J. E., Partyka, G., Pawson, S., Putman, W., Rienecker, M., Schubert, S. D., Sienkiewicz, M. and Zhao, B.: The Modern-Era Retrospective Analysis for Research and Applications, Version 2 (MERRA-2), J Clim, **Volume 30**(Iss 13): 5419-5454, 10.1175/jcli-d-16-0758.1, 2017.

Herrmann, H., Schaefer, T., Tilgner, A., Styler, S. A., Weller, C., Teich, M. and Otto, T.: Tropospheric aqueous-phase chemistry: kinetics, mechanisms, and its coupling to a changing gas phase, Chem Rev, **115**(10): 4259-4334, 10.1021/cr500447k, 2015.

Li, J., Wang, X., Chen, J., Zhu, C., Li, W., Li, C., Liu, L., Xu, C., Wen, L., Xue, L., Wang, W., Ding, A. and Herrmann, H.: Chemical composition and droplet size distribution of cloud at the summit of Mount Tai, China, Atmos. Chem. Phys., **17**(16): 9885-9896, 10.5194/acp-17-9885-2017, 2017.

Ma, T., Furutani, H., Duan, F., Kimoto, T., Jiang, J., Zhang, Q., Xu, X., Wang, Y., Gao, J., Geng, G., Li, M., Song, S., Ma, Y., Che, F., Wang, J., Zhu, L., Huang, T., Toyoda, M. and He, K.: Contribution of hydroxymethanesulfonate (HMS) to severe winter haze in the North China Plain, Atmospheric Chemistry and Physics, **20**(10): 5887-5897, 10.5194/acp-20-5887-2020, 2020.

Shah, V., Jacob, D. J., Moch, J. M., Wang, X. and Zhai, S.: Global modeling of cloud water acidity, precipitation acidity, and acid inputs to ecosystems, Atmos. Chem. Phys., **20**(20): 12223-12245, 10.5194/acp-20-12223-2020, 2020.

Song, S., Gao, M., Xu, W., Sun, Y., Worsnop, D. R., Jayne, J. T., Zhang, Y., Zhu, L., Li, M., Zhou, Z., Cheng, C., Lv, Y., Wang, Y., Peng, W., Xu, X., Lin, N., Wang, Y., Wang, S., Munger, J. W., Jacob, D. J. and McElroy, M. B.: Possible heterogeneous chemistry of hydroxymethanesulfonate (HMS) in northern China winter haze, Atmospheric Chemistry and Physics, **19**(2): 1357-1371, 10.5194/acp-19-1357-2019, 2019.

Wagner, V., Schiller, C. and Fischer, H.: Formaldehyde measurements in the marine boundary layer of the Indian Ocean during the 1999 INDOEX cruise of the R/V Ronald H. Brown, Journal of Geophysical Research: Atmospheres, **106**(D22): 28529-28538, 10.1029/2000jd900825, 2001.

Wang, H., Li, J., Wu, T., Ma, T., Wei, L., Zhang, H., Yang, X., Munger, J. W., Duan, F. K., Zhang, Y., Feng, Y., Zhang, Q., Sun, Y., Fu, P., McElroy, M. B. and Song, S.: Model Simulations and Predictions of Hydroxymethanesulfonate (HMS) in the Beijing-Tianjin-Hebei Region,

China: Roles of Aqueous Aerosols and Atmospheric Acidity, Environ Sci Technol, **58**(3): 1589-1600, 10.1021/acs.est.3c07306, 2024.

Wang, T., Liu, M., Liu, M., Song, Y., Xu, Z., Shang, F., Huang, X., Liao, W., Wang, W., Ge, M., Cao, J., Hu, J., Tang, G., Pan, Y., Hu, M. and Zhu, T.: Sulfate Formation Apportionment during Winter Haze Events in North China, Environmental Science & Technology, **56**(12): 7771-7778, 10.1021/acs.est.2c02533, 2022.

Wei, L., Fu, P., Chen, X., An, N., Yue, S., Ren, H., Zhao, W., Xie, Q., Sun, Y., Zhu, Q.-F., Wang, Z. and Feng, Y.-Q.: Quantitative Determination of Hydroxymethanesulfonate (HMS) Using Ion Chromatography and UHPLC-LTQ-Orbitrap Mass Spectrometry: A Missing Source of Sulfur during Haze Episodes in Beijing, Environmental Science & Technology Letters, **7**(10): 701-707, 10.1021/acs.estlett.0c00528, 2020.

Williams, A. G., Chambers, S. D., Conen, F., Reimann, S., Hill, M., Griffiths, A. D. and Crawford, J.: Radon as a tracer of atmospheric influences on traffic-related air pollution in a small inland city, Tellus B: Chemical and Physical Meteorology, **68**(1), 10.3402/tellusb.v68.30967, 2016.